# Walking the Schrödinger Bridge: A Direct Trajectory for Text-to-3D Generation

**Ziying Li**
Zhejiang University
emmaleee@zju.edu.cn

**Xuequan Lu**
University of Western Australia
bruce.lu@uwa.edu.au

**Xinkui Zhao***
Zhejiang University
zhaoxinkui@zju.edu.cn

**Guanjie Cheng**
Zhejiang University
chengguanjie@zju.edu.cn

**Shuiguang Deng**
Zhejiang University
dengsg@zju.edu.cn

**Jianwei Yin**
Zhejiang University
zjuyjw@cs.zju.edu.cn

https://github.com/emmaleee789/TraCe.git

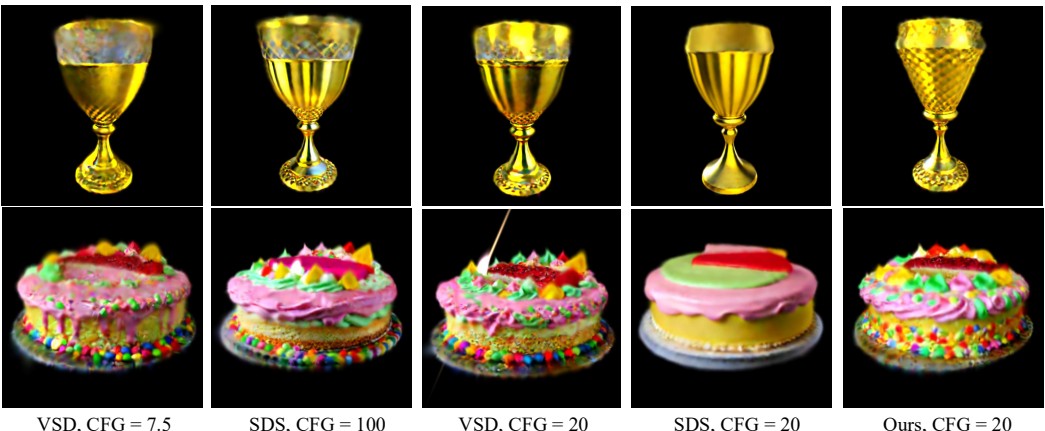

| VSD, CFG = 7.5 | SDS, CFG = 100 | VSD, CFG = 20 | SDS, CFG = 20 | Ours, CFG = 20 |

Figure 1: From left to right: (a) Standard VSD [45] (CFG = 7.5, CFG: Classifier-free Guidance); (b) Standard SDS [35]; (CFG = 100); (c) VSD [45] (CFG = 20); (d) SDS [35] (CFG = 20); (e) Ours (CFG = 20). VSD with CFG = 7.5 and CFG = 20 both yield low-quality results. Standard SDS yields artifacts (e.g., over-smoothing) with high CFG, and SDS with low CFG yields low-quality results. Our method generates high-quality and high-fidelity results with a fair CFG value.

## Abstract

Recent advancements in optimization-based text-to-3D generation heavily rely on distilling knowledge from pre-trained text-to-image diffusion models using techniques like Score Distillation Sampling (SDS), which often introduce artifacts such as over-saturation and over-smoothing into the generated 3D assets. In this paper, we address this essential problem by formulating the generation process as learning an optimal, direct transport trajectory between the distribution of the current rendering and the desired target distribution, thereby enabling high-quality generation with smaller Classifier-free Guidance (CFG) values. At first, we theoretically establish SDS as a simplified instance of the Schrödinger Bridge framework. We prove that SDS employs the reverse process of an Schrödinger

---

*Corresponding author. zhaoxinkui@zju.edu.cn

39th Conference on Neural Information Processing Systems (NeurIPS 2025).

Bridge, which, under specific conditions (e.g., a Gaussian noise as one end), collapses to SDS's score function of the pre-trained diffusion model. Based upon this, we introduce Trajectory-Centric Distillation (TraCe), a novel text-to-3D generation framework, which reformulates the mathematically trackable framework of Schrödinger Bridge to explicitly construct a diffusion bridge from the current rendering to its text-conditioned, denoised target, and trains a LoRA-adapted model on this trajectory's score dynamics for robust 3D optimization. Comprehensive experiments demonstrate that TraCe consistently achieves superior quality and fidelity to state-of-the-art techniques.

# 1 Introduction

Generating three-dimensional content directly from textual descriptions has recently attracted intensive attentions in the research community. Recent methods leveraging explicit 3D representations like Gaussian Splatting have significantly accelerated the generation process [25, 3]. Despite the advancements, it remains a key bottleneck that the quality and fidelity of generated 3D assets often lag behind their 2D counterparts. This limitation is frequently attributed to the scarcity of large-scale, high-quality 3D datasets required for direct supervised training [27, 28, 10].

To bridge this gap, many state-of-the-art text-to-3D methods employ optimization strategies guided by powerful, pre-trained 2D text-to-image (T2I) diffusion models [36]. Score Distillation Sampling (SDS) [35] has become the cornerstone paradigms. SDS leverages powerful pre-trained 2D text-to-image diffusion models to guide the optimization of 3D representations. Nevertheless, the standard SDS approach typically requires high values for Classifier-Free Guidance (CFG) [13] to achieve strong text alignment [35, 47, 4, 24, 49]. This reliance on high CFG values is often problematic, leading to visual artifacts such as over-saturation [37] and over-smoothing [23] in the generated 3D assets. Recognizing these issues, several variants of SDS have been proposed recently [45, 29, 17, 44, 48, 11, 6]. However, these SDS-based methods, including the recent variants, face persistent challenges. Firstly, as analyzed in recent studies [45, 1, 24], SDS and its variants fundamentally operate by matching the gradient direction predicted by the T2I model. While differing in their specific source and target choices for computing this gradient, they all rely on score estimates derived from the T2I backbone. These score estimates, however, can be noisy and are not guaranteed to represent an optimal direction for 3D optimization (shown in Figure 2b), potentially causing unexpected artifacts. Secondly, variants designed to operate effectively at lower CFG values (e.g., CFG=7.5), such as Score Distillation via Inversion (SDI) [29] or Variational Score Distillation (VSD) [45], have shown limited success when applied to optimizing certain popular 3D representations like 3D Gaussian Splatting (3DGS), often yielding less-desired results (shown in Figure 1).

The aforementioned analysis underscores the limitations of existing approaches and highlights the urgent need of a more robust optimization framework for text-to-3D generation, one that does not solely rely on potentially noisy score matching or operate under restrictive guidance conditions. In this paper, we first provide a theoretical insight by establishing that SDS can be understood as a simplified instance of the Schrödinger Bridge framework [39]. We demonstrate (Section 4.1) that SDS implicitly employs the reverse process of an Schrödinger Bridge, which, under specific conditions such as Gaussian noise distribution at one endpoint, effectively collapses to utilizing the score function of the pre-trained diffusion model. This perspective not only clarifies the underlying dynamics of SDS but also illuminates pathways for more principled trajectory design. Based upon this reformulation, we introduce **Tra**jectory-**Ce**ntric Distillation (TraCe), a novel text-to-3D generation framework. TraCe formulates the mathematically tractable framework of Schrödinger Bridges [26, 26] to explicitly construct and learn a diffusion bridge for text-to-3D generation. This bridge connects the current rendering ($X_1$) to its text-conditioned, denoised target ($X_0^{\text{pred}}$), thereby defining a more stable and direct optimization trajectory (visualization in Figure 2a). TraCe then employs Low-Rank Adaptation (LoRA) [14] to fine-tune the T2I diffusion model specifically for navigating this constructed bridge, enabling it to precisely learn the score dynamics required for robust 3D optimization along this optimal trajectory towards the target distribution.

Our proposed TraCe framework, which operationalizes the direct transport path via Schrödinger Bridges, is rigorously evaluated. Extensive experiments demonstrate that this approach yields high-fidelity 3D assets with strong adherence to textual descriptions (Figure 4 and Table 1). The results consistently showcase TraCe's capacity to achieve superior visual quality and semantic coherence

in generated content (Figure 4 and Supplementary), highlighting the efficacy of our theoretically grounded direct trajectory optimization for text-to-3D generation.

In summary, our contributions are:

- We establish a novel theoretical connection, demonstrating that SDS can be precisely understood as a special case of the Schrödinger Bridge framework. This reformulation clarifies the underlying transport dynamics implicitly leveraged by SDS.

- We introduce Trajectory-Centric Distillation (TraCe), a new text-to-3D generation framework. TraCe explicitly learns an optimal transport path, guided by a tractable Schrödinger Bridge formulation, between the current 3D model's rendering and a dynamically estimated, text-aligned target view. This is achieved by constructing and sampling along this explicit diffusion bridge, enabling more direct and stable 3D optimization.

- Experiments demonstrate that our TraCe achieves high-quality 3D generation, surpassing current state-of-the-art techniques. TraCe exhibits enhanced robustness, particularly excelling in challenging low CFG values where the performance of existing methods typically degrades.

## 2 Related Work

**Distilling 2D into 3D.** Leveraging large-scale, pre-trained text-to-image (T2I) diffusion models [36] as priors has become a prominent technique for generation tasks in data-scarce domains, such as text-to-3D generation. SDS [35] is a seminal approach in this direction, enabling optimization of parametric representations (e.g., Neural Radiance Fields) by distilling knowledge from a 2D diffusion model. To achieve plausible results, it frequently necessitates high Classifier-Free Guidance (CFG) weights [35, 47], which can further exacerbate these issues. However, standard SDS is often susceptible to visual artifacts such as over-saturation [37] and over-smoothing [23]. Moreover, the SDS objective itself, while empirically effective, does not strictly correspond to the gradient of a well-defined probability distribution of the 3D parameters [45, 1, 24], potentially leading to suboptimal optimization paths [17, 44, 29, 48]. To address these limitations, several variants have been proposed. For instance, methods like Variational Score Distillation (VSD) [45] and Classifier Score Distillation (CSD) [48] explore alternative gradient formulations to better approximate the optimization process from source distribution towards target distribution. Other approaches like Score Distillation via Inversion (SDI) [29] tries to better approximate the noise instead of using pure Gaussian noise. These variants can be understood through the lens of approximating an optimal transport path between the current image distribution (source) and the target natural image distribution, and from this perspective, a key difference between these methods lies in how they approximate the score of the source and target distributions [30]. For instance, SDS approximates it using the unconditional score, while VSD attempts a more direct approximation by fine-tuning a LoRA adapter on the current renderings. While these methods offer valuable contributions towards reducing the source distribution mismatch artifacts, they fundamentally rely on adapting gradients derived from pre-trained T2I models. This forces the optimization process to cope with score functions optimized for 2D image generation, which is inherently not optimal for tasks like 3D generation due to the domain gap and differences. Our work differs greatly from these approaches. We establish a novel theoretical connection, demonstrating that SDS can be precisely understood as a specific instantiation of the Schrödinger Bridge framework. This reformulation clarifies the underlying transport dynamics implicitly leveraged by SDS. Built upon this insight, we introduce a method that explicitly constructs and learns a more direct and stable optimization trajectory by framing the process as a tractable Schrödinger Bridge between the current rendering and an estimated text-aligned target, thereby enhancing both the fidelity and robustness of text-to-3D generation.

**Diffusion Models and Schrödinger Bridges.** Diffusion models (DMs) [12], also known as Score-based Generative Models (SGMs) [40, 42], have emerged as a dominant class of deep generative techniques, achieving state-of-the-art performance in synthesizing high-fidelity data across various domains, notably images [40, 12, 42, 9]. These models typically define a forward diffusion process, often formulated as a stochastic differential equation (SDE), that gradually corrupts data samples into a simple prior distribution, usually Gaussian noise. A neural network is then trained, often via score-matching objectives [16, 43, 42], to approximate the score function (gradient of the log density) of the perturbed data distributions. This learned score function parameterizes a reverse-time

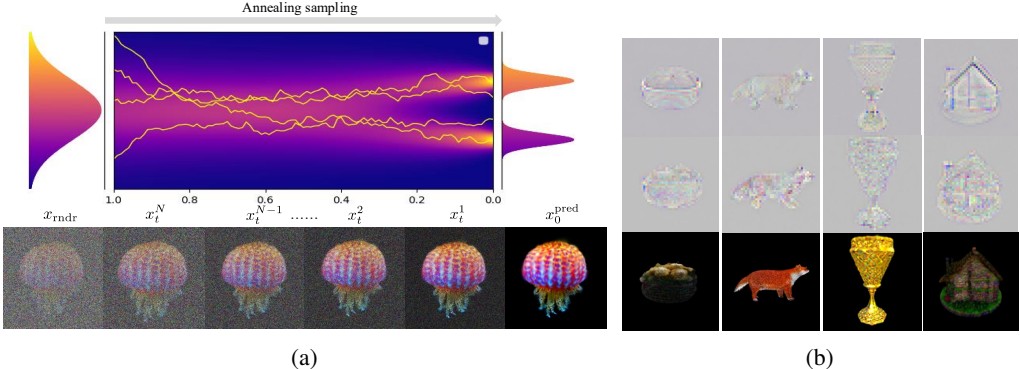

(a)                 (b)

Figure 2: **Left: Schrödinger Bridge Visualization and Samples.** Top: Probability flow of the bridge from current rendering ($x_{\mathrm{rndr}}$) to the predicted target ($x_0^{\mathrm{pred}}$) distribution. Bottom: Corresponding image samples, showing the current rendering, intermediate bridge samples ($x_t^i$), and the final predicted target. **Right: Gradient and Intermediate Rendering Comparison.** The first row shows TraCe gradients, the second shows SDS gradients, and the third shows rendered images of the 3D models that have not finished generation. Note the reduced artifacts and potentially more coherent structure in the TraCe gradients and intermediate renderings.

SDE that transforms samples from the prior back into data samples. While being extremely successful, this standard paradigm typically relies on initiating the generative process from unstructured noise. The Schrödinger Bridge problem provides a more general theoretical framework, originating from statistical physics [38, 39] and connected to entropy-regularized optimal transport [21, 5] and stochastic control [7, 34]. It aims to find the most likely stochastic evolution between two specified arbitrary distributions, $P_A$ and $P_B$, rather than being restricted to a noise prior. This offers the potential to learn direct transformations between complex data manifolds. Attempts have been made to apply Schrödinger Bridge concepts to text-to-3D generation. For instance, [30] proposes a naive approach to direct Schrödinger Bridge formulation between current renderings and target images guided by text prompts, though this requires an initial stage involving standard SDS. Another approach, DreamFlow [20], proposes to approximate the backward Schrödinger Bridge dynamics between current renderings and target images by simply repurposing a fine-tuned text-to-image model, a heuristic potentially deviating from the true underlying Schrödinger Bridge process. We critically advance text-to-3D generation by establishing the precise theoretical relationship between SDS and Schrödinger Bridges. This foundational insight is then exploited to develop a principled methodology for direct distributional transport, enabling the construction of trajectories towards text-aligned target distributions.

## 3 Preliminaries

**Score-based Generative Model (SGM) and Schrödinger Bridge.** Score-based Generative Models (SGM) [40, 42] learn to generate data by reversing a predefined forward diffusion process. This process gradually transforms data $X_0 \sim p_{\mathcal{A}}$ into noise $X_1 \approx \mathcal{N}(0, I)$ and is often governed by a forward stochastic differential equation (SDE). Generation then proceeds by simulating the corresponding reverse-time SDE [2], starting from $X_1$ and integrating backward to $t = 0$. The forward and reverse SDEs are given by:

$$dX_t = f_t(X_t)dt + g_t dW_t \text{ (forward)}$$
$$dX_t = \left[ f_t(X_t) - g_t^2 \nabla_{X_t} \log p(X_t, t) \right] dt + g_t d\bar{W}_t \text{ (backward)}$$

Here, $W_t$ (and $\bar{W}_t$) is a standard Wiener process, and $g_t$ represents the time-dependent diffusion coefficient. The central part of this reversal is the score function $\nabla_{X_t} \log p(X_t, t)$, which is unknown and approximated using a time-conditioned neural network $s_\psi(X_t, t)$ (or an equivalent noise predictor $\epsilon_\psi(X_t, t)$). This network is trained using score-matching objectives [43, 42] on pairs $(X_0, X_t)$ sampled from the forward process. Sampling is performed by numerically integrating the reverse SDE using solvers like DDPM [12] or DDIM [41].

The Schrödinger Bridge problem [39, 21] offers a generalization of SGMs to learn nonlinear diffusion processes between two arbitrary distributions, $X_0 \sim p_{\mathcal{A}}$ and $X_1 \sim p_{\mathcal{B}}$. It seeks the most likely stochastic evolution connecting these boundary distributions, described by a pair of forward and backward SDEs:

$$dX_t = \left[ f_t(X_t) + \beta_t \nabla \Psi(X_t, t) \right] dt + \sqrt{\beta_t} dW_t \text{ (forward)}$$

$$dX_t = \left[ f_t(X_t) - \beta_t \nabla \hat{\Psi}(X_t, t) \right] dt + \sqrt{\beta_t} d\bar{W}_t \text{ (backward)}$$

where $\Psi(x, t)$ and $\hat{\Psi}(x, t)$ are non-negative functions known as Schrödinger factors, determined by coupled partial differential equations with boundary conditions $\Psi(x, 0)\hat{\Psi}(x, 0) = p_{\mathcal{A}}(x)$ and $\Psi(x, 1)\hat{\Psi}(x, 1) = p_{\mathcal{B}}(x)$. The forward and backward processes induce the same marginal density $q(x, t)$ at any time $t \in [0, 1]$, satisfying Nelson's duality $\Psi(x, t)\hat{\Psi}(x, t) = q(x, t)$ [33]. Notably, SGM is a special case where $p_{\mathcal{B}} \approx \mathcal{N}(0, I)$ and $\Psi(x, t) \approx 1$, causing the forward drift modification to vanish and $\hat{\Psi}(x, t) \approx q(x, t)$, recovering the score function in the reverse SDE.

**Score Distillation Sampling (SDS).**  Score Distillation Sampling (SDS) [35] enables generating 3D assets by leveraging powerful pre-trained 2D text-to-image diffusion models [36], bypassing the need for large-scale 3D datasets. It optimizes the parameters $\theta$ of a differentiable 3D representation, such as NeRF [31], InstantNGP [32], or 3D Gaussian Splatting (3DGS) [18], using gradients derived from the diffusion model. In this work, we adopt 3DGS primarily for its rapid generation capabilities and high-fidelity visual output.

The core mechanism of SDS involves repeatedly rendering the 3D model from different views $c$ ($x = g(\theta, c)$), adding noise to the rendering $x(t)$, and using the 2D diffusion model's score estimate (denoising prediction $\epsilon_{\text{pred}}$) to guide the optimization of $\theta$. Formally, the gradient is computed as

$$\nabla_\theta \mathcal{L}_{\text{SDS}}(\theta) = \mathbb{E}_{t, \epsilon, c} \left[ w(t) \left( \epsilon_{\text{pred}} - \epsilon_{\text{noise}} \right) \frac{\partial x_{\text{rndr}}}{\partial \theta} \right] \tag{1}$$

where $w(t)$ is a weighting factor and the term $(\epsilon_{\text{pred}} - \epsilon_{\text{noise}})$ provides the guidance signal. While SDS can be intuitively understood as moving renderings towards higher-density regions according to the 2D prior or formally interpreted via probability density distillation, the exact nature of its gradient signal is debated [17, 48, 44, 1, 45]. Practically, SDS often requires high classifier-free guidance (CFG) values, which can sometimes lead to artifacts like oversaturation or blur [17, 29, 46, 19, 37, 22]. Furthermore, the strategies that employ lower CFG values, for instance, methods explored in text-to-NeRF [45, 29], have demonstrated limitations when directly applied to the generation of 3D assets with Gaussian Splatting (Figure 4). Recent efforts such as LucidDreamer [23] have investigated text-to-3DGS under low CFG conditions; however, this direction currently faces trade-offs, including prolonged optimization durations (over 5000 iterations) and limitations in the attainable visual quality (Figure 4). Our work builds upon SDS by mitigating these issues through deriving a more direct and tractable optimization path, formulating Schrödinger Bridges to guide the generation process for achieving greater fidelity with lower CFG values.

## 4   Method

The preceding analysis of existing methods like Score Distillation Sampling (SDS) raise a natural question: can a principled framework be developed to define a direct, optimal transformation trajectory where both its source and target ends are explicitly and robustly aligned with the desired true distributions? Addressing this challenge—by establishing explicit control over the distributional endpoints of the generative trajectory, rather than relying on unstructured priors (e.g., a Gaussian noise)—is crucial for enhancing the fidelity and control of generative outcomes. To this end, we exploit the theoretical underpinnings of the Schrödinger Bridge problem, particularly its tractable formulations [26, 8], which provide a robust mechanism for learning direct, optimal transport paths between specified distributions. Our methodological contribution unfolds in two stages: first, we theoretically establish that standard SDS is indeed a special case of the Schrödinger Bridge framework, thereby providing a new perspective on its operation (Section 4.1). Second, building upon this insight, we propose a novel optimization algorithm grounded in tractable Schrödinger Bridge principles, to achieve improved distributional alignment throughout the generative process (Section 4.2).

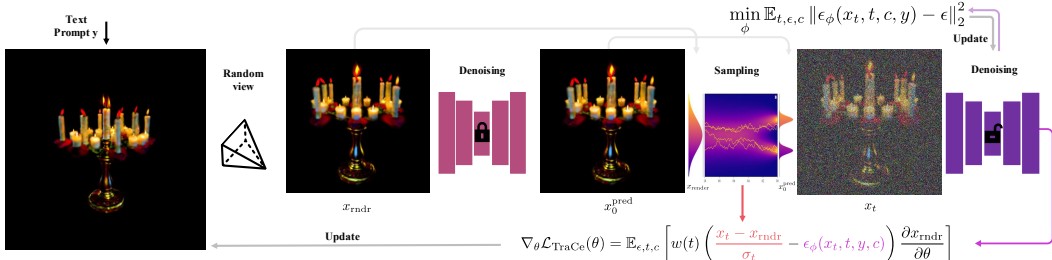

$$\min_{\phi} \mathbb{E}_{t,\epsilon,c} \|\epsilon_\phi(x_t,t,c,y) - \epsilon\|_2^2$$

$$\nabla_\theta \mathcal{L}_{\text{TraCe}}(\theta) = \mathbb{E}_{\epsilon,t,c}\left[ w(t)\left( \frac{x_t - x_{\text{rndr}}}{\sigma_t} - \epsilon_\phi(x_t,t,y,c) \right) \frac{\partial x_{\text{rndr}}}{\partial \theta} \right]$$

Figure 3: **Overview of Trajectory-Centric Distillation (TraCe).** Our TraCe optimizes 3D parameters $\theta$ by computing a distillation gradient with a LoRA-adapted 2D diffusion model, $\epsilon_\phi$. Given a text prompt $y$ and camera parameters $c$, (1) the current 3D model is rendered in a random view to produce $x_{\text{rndr}}$. (2) An ideal target view $x_0^{\text{pred}}$ is estimated from $x_{\text{rndr}}$ using a pre-trained diffusion model $\epsilon_{\text{pretrain}}$ via one-step denoising. (3) An intermediate latent $x_t$ is sampled from the analytic bridge posterior $q(x_t \mid x_0^{\text{pred}}, x_{\text{rndr}})$ at time $t$. (4) The LoRA model $\epsilon_\phi$ predicts the noise for $x_t$, and the difference between this prediction and the target noise is computed. (5) This difference directs the calculation of the TraCe gradient $\nabla_\theta \mathcal{L}_{\text{TraCe}}$, and drives the update of LoRA parameters $\phi$.

## 4.1 Score Distillation Sampling as a Special Case of Schrödinger Bridges

In this section, we reformulate the SDS objective by examining its core guidance principles, and show it employs a simplified form of the backward dynamics found in the Schrödinger Bridge framework.

As established in Section 3, a Score-based Generative Model (SGM) aligns with a special configuration of the Schrödinger Bridge problem. This occurs when the Schrödinger Bridge's distribution $P_B$ at $t=1$ is Gaussian noise ($P_B \sim \mathcal{N}(0,I)$) and its forward Schrödinger factor $\Psi(x,t) \approx 1$. Under these conditions, the term $g_t^2 \nabla_{X_t} \log \Psi(X_t,t)$ in the forward Schrödinger Bridge SDE vanishes, causing the forward Schrödinger Bridge dynamics to become identical to the SGM's standard diffusion process. Consequently, the marginal densities $q(X_t,t)$ of this particular Schrödinger Bridge are equivalent to the SGM's noisy marginals $p(X_t,t)$.

The crucial step in linking the Schrödinger Bridge and SGM reverse processes from a score perspective lies in Nelson's duality, $\Psi(X_t,t)\hat{\Psi}(X_t,t) = q(X_t,t)$. Given $\Psi(X_t,t) \approx 1$ and $q(X_t,t) = p(X_t,t)$ for this specific Schrödinger Bridge, the duality simplifies to:

$$1 \cdot \hat{\Psi}(X_t,t) \approx p(X_t,t) \implies \hat{\Psi}(X_t,t) \approx p(X_t,t) \tag{2}$$

This directly implies that the score term in the general Schrödinger Bridge backward SDE, $-\nabla_{X_t} \log \hat{\Psi}(X_t,t)$, becomes $-\nabla_{X_t} \log p(X_t,t)$. This is precisely the score approximated by the learned network $s_\psi(X_t,t)$ (or its equivalent noise predictor $\epsilon(X_t,t)$) in an SGM.

SDS utilizes this learned score $s_\psi(X_t,t)$ from a pre-trained SGM to guide the optimization of a differentiable generator $g(\theta)$. The update for $g(\theta)$ is fundamentally derived from $s_\phi(X_t,t)$, aiming to make the generated samples $x_0 = g(\theta)$ consistent with the data manifold learned by the SGM.

Therefore, from a score gradient perspective:

- SDS operates using the score function $s_\psi(X_t,t)$ learned by an SGM.

- The derivation above shows that $s_\psi(X_t,t)$ (approximating $\nabla_{X_t} \log p(X_t,t)$) is equivalent to the score $-\nabla_{X_t} \log \hat{\Psi}(X_t,t)$ of a Schrödinger Bridge under the specific conditions that reduce the Schrödinger Bridge to an SGM.

**Remark.** In essence, SDS leverages a score gradient that is equivalent to the score function governing the reverse dynamics of the canonical Schrödinger Bridge implicit in any SGM. While general Schrödinger Bridges can offer more complex dynamics, SDS employs the score from this specific, simplified Schrödinger Bridge structure. Thus, the SDS mechanism represents an application of principles governing a special case of Schrödinger Bridges, distinguished by its reliance on the SGM-derived score $s_\psi$.

## 4.2  Trajectory-Centric Distillation

To optimize the 3D model parameters $\theta$ such that current renderings $x_{\text{rndr}} = g(\theta, c)$ align with a target text description $y$, we propose a novel method, namely Trajectory-Centric Distillation (TraCe). This method leverages a 2D diffusion model, adapted with LoRA parameters $\phi$ denoted as $\epsilon_\phi$, to provide a guiding gradient $\nabla_\theta \mathcal{L}_{\text{TraCe}}(\theta)$. The core idea is to conceptualize a diffusion bridge between the current rendering and an estimated ideal target image.

**Constructing the Diffusion Bridge for Trajectory Guidance.**  At each optimization step for $\theta$, we construct a specific diffusion bridge instance defined by two endpoints:

1. **Initial Bridge Endpoint** ($X_1 \leftarrow x_{\text{rndr}}$): The current rendering $x_{\text{rndr}} = g(\theta, c)$ serves as the starting point of the reverse diffusion trajectory we aim to learn. In the context of our bridge, this is treated as the "noisier" state at bridge time $t = 1$.

2. **Target Bridge Endpoint** ($X_0 \leftarrow x_0^{\text{pred}}$): An estimated ideal target view $x_0^{\text{pred}}$ acts as the desired endpoint at bridge time $t = 0$. This target is dynamically obtained by performing one-step denoising on $x_{\text{rndr}}$ using a pre-trained text-to-image model $\epsilon_{\text{pretrain}}$ [20], conditioned on the text prompt $y$: $x_0^{\text{pred}} = \left(x_{\text{rndr}} - \sqrt{1 - \bar{\alpha}_{t'}}\, \epsilon_{\text{pretrain}}(x_{\text{rndr}}, t', y)\right) / \sqrt{\bar{\alpha}_{t'}}$, where $\bar{\alpha}_{t'}$ is from the noise schedule of $\epsilon_{\text{pretrain}}$.

With these two endpoints, $x_0^{\text{pred}}$ and $x_{\text{rndr}}$, established, we then sample an intermediate latent state $x_t$ along the conceptual bridge. For a sampled time $t \in [0.02, 0.5]$, following the tractable formulation of Schrödinger Bridges [26], $x_t$ is drawn from the analytically known conditional distribution $x_t \sim q(x_t | x_0^{\text{pred}}, x_{\text{rndr}}) = \mathcal{N}(x_t; \boldsymbol{\mu}_t, \Sigma_t I)$, where the mean $\boldsymbol{\mu}_t = \gamma_t x_0^{\text{pred}} + (1 - \gamma_t) x_{\text{rndr}}$ is an interpolation between the target image and current rendering, and $\Sigma_t = \sigma_t^2 \bar{\sigma}_t^2 / (\sigma_t^2 + \bar{\sigma}_t^2)$ is the bridge variance. The coefficient $\gamma_t = \bar{\sigma}_t^2 / (\sigma_t^2 + \bar{\sigma}_t^2)$, and $\sigma_t^2 = \int_0^t \beta_\tau d\tau$, $\bar{\sigma}_t^2 = \int_t^1 \beta_\tau d\tau$ are accumulated variances from a noise schedule $\beta_t$ specific to this bridge construction. This $x_t$ represents a state on a direct trajectory from $x_0^{\text{pred}}$ being progressively "noised" towards $x_{\text{rndr}}$ (or equivalently, $x_{\text{rndr}}$ being progressively "denoised" towards $x_0^{\text{pred}}$ along this trajectory).

**Optimizing $\theta$ via the Bridge Trajectory.**  We then optimize $\theta$ using the LoRA-adapted model $\epsilon_\phi(x_t, t, y, c)$, which is trained to predict the noise that would take $x_t$ towards $x_0^{\text{pred}}$. The objective for $\theta$ utilizes $\epsilon_\phi$ to measure the consistency of $x_t$ with respect to $x_{\text{rndr}}$ along this bridge:

$$\nabla_\theta \mathcal{L}_{\text{TraCe}}(\theta) = \mathbb{E}_{\epsilon, t, c} \left[ w(t) \left( \epsilon_\phi(x_t, t, y, c) - \frac{x_t - x_{\text{rndr}}}{\sigma_t} \right) \left( \underbrace{\frac{\partial x_0^{\text{pred}}(x_{\text{rndr}}, t, y)}{\partial x_t} \frac{\partial x_t}{\partial x_{\text{rndr}}} + 1}_{\text{U-net Jacobian}} \right) \frac{\partial x_{\text{rndr}}}{\partial \theta} \right]$$

(3)

where $x_{\text{rndr}} = g(\theta, c)$, $t \sim \mathcal{U}[0.02, 0.5]$, and $y$ is the text prompt. The term $x_t$ is sampled from $q(x_t \mid x_0^{\text{pred}}, x_{\text{rndr}})$ as defined above, and $\sigma_t = \sqrt{\int_0^t \beta_\tau d\tau}$ from the bridge's noise schedule. Following the convention of SDS, we omit the U-Net Jacobian term $\left( \frac{\partial x_0^{\text{pred}}(\dots)}{\partial x_t} \frac{\partial x_t}{\partial x_{\text{rndr}}} + 1 \right)$ for effective training, as it can be treated as a learnable or constant factor absorbed by $w(t)$. Thus, we have:

$$\nabla_\theta \mathcal{L}_{\text{TraCe}}(\theta) = \mathbb{E}_{\epsilon, t, c} \left[ w(t) \left( \epsilon_\phi(x_t, t, y, c) - \frac{x_t - x_{\text{rndr}}}{\sigma_t} \right) \frac{\partial x_{\text{rndr}}}{\partial \theta} \right]$$

(4)

**Scheduled $t$-Sampling for Schrödinger Bridges Interpolation.**  For sampling the intermediate state $x_t$ in our TraCe objective (Eq. (4)), which dictates the characteristics of $x_t \sim q(x_t \mid x_0^{\text{pred}}, x_{\text{rndr}})$, we adopt a $t$-annealing strategy, similar to the approach proposed in [15]. Throughout the optimization of $\theta$, the time parameter $t$ is progressively decreased from an initial value near $0.5$ towards $0.02$. This common annealing technique gradually shifts the focus of the Schrödinger Bridge interpolation from broader states towards those more proximate to the estimated ideal target $x_0^{\text{pred}}$, aiding the progressive refinement of the rendered output $g(\theta, c)$.

# 5 Experiments

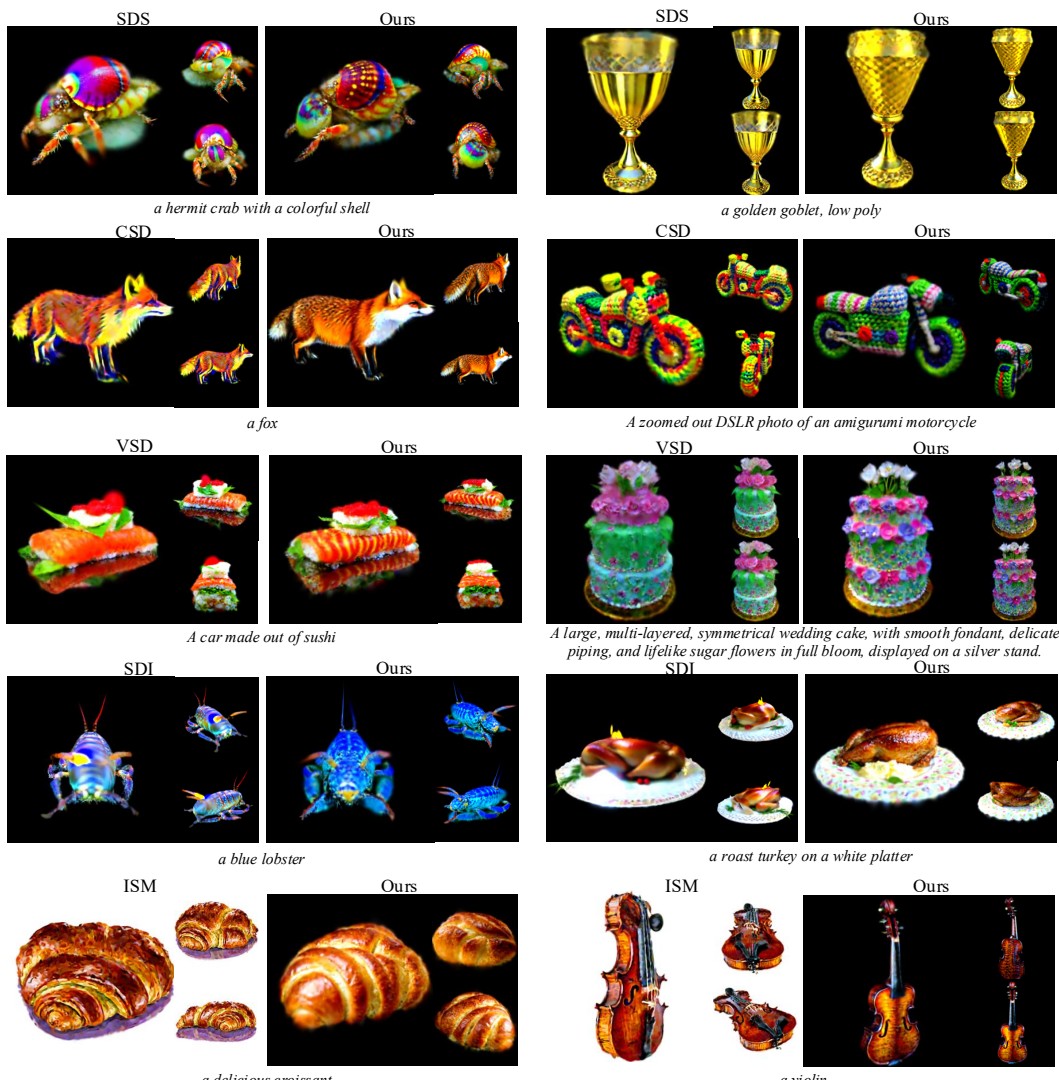

Figure 4: **Qualitative comparisons.** We present visual examples with the same text prompt.

**Implementation Details.** We choose recent state-of-the-art (SOTA) text-to-3D approaches for comparison: NeRF-based methods, such as Classifier Score Distillation (CSD) [48], ProlificDreamer (VSD) [45], and Score Distillation via Inversion (SDI) [29], and 3DGS-based methods like GaussianDreamer (SDS) [47] and LucidDreamer (ISM) [23]. Please see more details and experiments in Supplementary.

**Qualitative Comparisons.** Figure 4 presents visual results for several challenging text prompts. Our approach demonstrates the ability to generate higher quality 3D assets compared to other SOTA methods. Compared to SDI [29], our method yields significantly improved texture fidelity. Outputs from CSD [48] often exhibit a characteristic yellowish hue and a less realistic, cartoon-like appearance, which TraCe avoids, producing more natural color rendition and photorealism. When compared against VSD [45], our model better interprets complex textural and stylistic prompts, accurately capturing the text's message and generating a more coherent content. Contrasting with SDS [47], our results exhibit superior sharpness and finer details in both geometry and texture, leading to more visually appealing and realistic outputs. While ISM [23] can produce coherent structures, its outputs often exhibit a stylized, painterly quality; in contrast, our TraCe generates

results with enhanced photorealism and more natural material appearance. These results demonstrate our method's effectiveness in generating detailed and accurate 3D geometry and appearance from the given text descriptions.

**Quantitative Comparison.** We quantitatively evaluate our TraCe against other methods using 83 distinct prompts from Dreamfusion online gallery[2] with 120 views per prompt. We benchmark generation quality using CLIP Score (%), GPTEval3D (Overall) (which leverages GPT-4o for evaluation), and ImageReward. CLIP Scores are evaluated with ViT-L/14, ViT-B/16, and ViT-B/32 backbones. We also assess computational efficiency via processing time (Time) and average peak VRAM (VRAM). As shown in Table 1, the proposed TraCe achieves state-of-the-art generation quality, securing top CLIP Scores across all ViT backbones, e.g., $69.2609 \pm 7.8366\%$ with ViT-L/14. Furthermore, TraCe demonstrates superior performance in advanced perception metrics, yielding the highest GPTEval3D score of 1028.03 and the most favorable (least negative) ImageReward score of $-0.2855 \pm 0.8909$, indicating enhanced aesthetic quality and semantic alignment. With an average processing time of 14 minutes and an average peak VRAM usage of 18741 MiB, TraCe offers high-fidelity generation with a compelling balance of qualitative performance, computational efficiency, and memory footprint.

Table 1: **Quantitative comparisons.** Comparison of different methods on CLIP Score, GPTEval3D Score, ImageReward Score, running time, and VRAM usage. We report mean and standard deviation across 83 prompts and 120 views.

| Method | CLIP Score (%) ↑ | | | GPTEval3D (Overall)↑ | ImageReward↑ | Time | VRAM |
|---|---|---|---|---|---|---|---|
| | ViT-L/14 | ViT-B/16 | ViT-B/32 | | | | |
| SDS [47] | 68.6146±7.9134 | 27.7049±3.7004 | 27.5561±3.5893 | 1018.09 | -0.4329±0.9125 | 10min | 18147MiB |
| CSD [48] | 68.0282±7.5093 | 27.0886±3.7342 | 26.5844±3.8703 | 983.04 | -0.6715±0.7482 | 11min | 19804MiB |
| VSD [45] | 67.2697±8.5573 | 27.0749±3.9675 | 26.9722±3.9563 | 1007.49 | -0.5330±0.8927 | 17min | 26473MiB |
| ISM [23] | 69.0093±10.2400 | 27.5460±3.6817 | 26.9822±3.5495 | 1012.37 | -0.3904±0.9503 | 20min | 10151MiB |
| SDI [29] | 63.0409±11.7841 | 25.6487±5.2540 | 25.5421±5.0903 | 971.98 | -0.8334±1.0391 | 10min | 16011MiB |
| **TraCe** | **69.2609±7.8366** | **27.9334±3.7382** | **27.7049±3.8671** | **1028.03** | **-0.2855±0.8909** | 14min | 18741MiB |

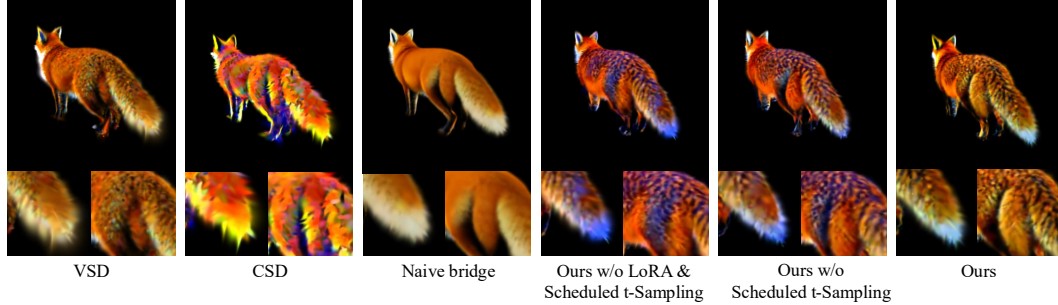

Figure 5: Ablation study on our framework.

**Ablation Study.** Figure 5 showcases the ablation study of our TraCe on a fox generation. VSD [45] and CSD [48] exhibit less-desired generation (e.g., missing details). The third column illustrates a naive Schrödinger Bridge approach [30] which attempts to bridge distributions defined by source and target prompts and results in a comparatively smoother, less detailed rendering. The fourth column shows TraCe without LoRA adaptation and without our scheduled $t$-sampling, where noticeable artifacts such as blue hues on the fur are apparent. Introducing LoRA but omitting the scheduled $t$-sampling (fifth column) mitigates some artifacts, yet color inconsistencies persist. Finally, our full TraCe method ("Ours")—supported by LoRA-adapted learning of its specific score dynamics and an annealed t-sampling schedule—generates significantly higher-fidelity details in the fur and tail, boosting overall realism compared to other methods (VSD, CSD) and ablated versions. These results highlight the role of our core Schrödinger Bridge formulation: it achieves superior final quality when augmented with these tailored learning components.

---

[2]https://dreamfusion3d.github.io/gallery.html

Table 2: ImageReward ablation over LoRA and scheduled $t$-sampling.

| Method Configuration | ImageReward ($\uparrow$) |
|---|---|
| LoRA off & scheduled $t$-sampling off | -0.4488 $\pm$ 0.9964 |
| LoRA off & scheduled $t$-sampling on | -0.3389 $\pm$ 0.9721 |
| LoRA on & scheduled $t$-sampling off | -0.4020 $\pm$ 1.0019 |
| LoRA on & scheduled $t$-sampling on (ours) | -0.2486 $\pm$ 0.8909 |

We perform an ablation study on our key components, LoRA adaptation and scheduled $t$-sampling, measuring quality with ImageReward (Table 2). Our full method (-0.2486) significantly outperforms the baseline (both off: -0.4488), as well as enabling only LoRA (-0.4020) or only scheduled $t$-sampling (-0.3389). The results confirm both components are crucial and demonstrate their strong synergistic effect.

**CFG value.** We investigate the impact of the CFG value on our TraCe, as illustrated in Figure 6 with two example objects. While very low CFG values (e.g., 5) yield reduced visual fidelity, TraCe produces high-quality, well-defined results starting at a CFG of approximately 15-20. The visual outcomes are stable and robust within the CFG 15-20 range. Beyond this, at higher CFG values (25-100), results remain largely consistent with minimal further improvement. This demonstrates TraCe's capability to effectively generate high-quality 3D assets at relatively low and stable CFG settings. Furthermore, TraCe enhanced visual quality is complemented by its robust CLIP score performance within a moderate CFG range (e.g., 10-30) relative to other compared methods, as detailed in Figure **??**.

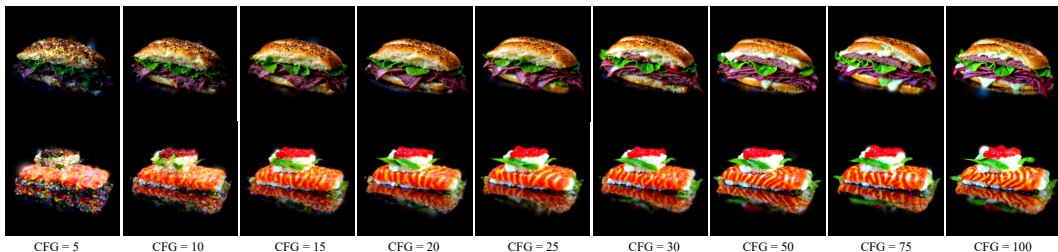

| CFG = 5 | CFG = 10 | CFG = 15 | CFG = 20 | CFG = 25 | CFG = 30 | CFG = 50 | CFG = 75 | CFG = 100 |

Figure 6: **Different CFG value and generated 3D assets.** Prompts are "an overstuffed pastrami sandwich" (top row), "a car made out of sushi" (bottom row).

# 6    Conclusion

We introduce Trajectory-Centric Distillation (TraCe), a novel text-to-3D generation framework. Our approach is rooted in a new theoretical understanding of SDS as a specific instance of the Schrödinger Bridge problem. The proposed TraCe explicitly constructs and learns a direct diffusion bridge between current renderings and text-conditioned targets, employing a LoRA-adapted diffusion model to accurately model the bridge's score dynamics. Comprehensive experiments demonstrate TraCe's state-of-the-art performance, yielding 3D assets with superior visual quality and fidelity, notably at lower and more stable Classifier-Free Guidance values than prior methods. These results underscore the benefits of our principled, direct optimization trajectory. We believe TraCe will offer new insights for text-to-3D generation, in terms of efficient and robust trajectory learning for generative models.

# 7    Acknowledgments

This work was supported in part by the National Science Foundation of China under Grants (62472375), and in part by the Major Program of National Natural Science Foundation of Zhejiang (LD24F020014, LD25F020002), and in part by the Zhejiang Pioneer (Jianbing) Project (2024C01032), and in part by the Ningbo Yongjiang Talent Programme(2023A-198-G).

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
