# OpenReview forum: "Walking the Schrödinger Bridge: A Direct Trajectory for Text-to-3D Generation"
_NeurIPS.cc/2025/Conference — NeurIPS 2025 poster_

### Official Review · Reviewer_ifTq · 2025-07-01

**Clarity:** 3
**Significance:** 2
**Originality:** 3
**Rating:** 4
**Confidence:** 5

**Summary:**

This paper introduces Trajectory-Centric Distillation (TraCe), a new framework for text-to-3D generation that reformulates the Score Distillation Sampling (SDS) approach under the Schrödinger Bridge framework. By utilizing LoRA adaptation and an annealed sampling schedule, TraCe mitigates over-saturation and over-smoothing artifacts that often occur with high classifier-free guidance (CFG) values in existing methods.

**Questions:**

1. The experimental setup raises concerns. In Table 1, the reported optimization time for VSD is questionable. In NeRF-based representations, VSD typically requires at least 2.5 hours per object, and even with Gaussian representations, it is unlikely for the training time to be shorter than that of ISM. Moreover, the default optimization steps for related methods are generally around 5000, whereas this work only uses 1700 steps, which may artificially reduce the reported optimization time. Even though incorporating LoRA may significantly increase training time, it is important to ensure that each model is fully converged before making fair comparisons.

2. The chosen evaluation metric (CLIP Score) does not align well with the stated motivation of improving texture quality and overall visual fidelity. CLIP primarily measures text-image alignment and cannot sufficiently evaluate texture quality. To support the claims more robustly, it would be more appropriate to incorporate human-aligned evaluation protocols, following GPT-4V(ision) or T^3Bench.

3. The paper lacks comparisons with some closely related and recent works, such as:
Consistent3D: Towards Consistent High-Fidelity Text-to-3D Generation with Deterministic Sampling Prior
Dreamer XL: Towards High-Resolution Text-to-3D Generation via Trajectory Score Matching

4. The ablation study lacks quantitative evaluation, making it difficult to rigorously assess the contribution of each component. Additionally, the distinction between the naive bridge and the proposed TraCe remains unclear. From my understanding, both approaches appear to rely on Equation (4) to update the 3D representation. Furthermore, the role of LoRA seems limited to color correction, yet incorporating LoRA significantly increases optimization time. This raises questions about whether the additional complexity and computational cost introduced by LoRA are justified by corresponding performance gains.

**Ethical Concerns:**

["NO or VERY MINOR ethics concerns only"]

**Final Justification:**

After the discussion, the major concern about the fairness of the comparison has been resolved. Although this work shows weaker visualizations, it provides better quantitative results. For this reason, I have raised the score.

**Limitations:**

This work lacks a discussion about limitations and failure cases.
- Please analyse failure cases.
- Sensitivity to LoRA fine-tuning stability and the risk of underfitting or mode collapse.

**Quality:**

2

**Strengths And Weaknesses:**

# Strengths
- The paper provides a theoretical analysis connecting SDS to Schrödinger Bridge, which offers a fresh perspective on optimization-based 3D generation.
- Mathematical descriptions, especially of the bridge construction and gradient computation, are clearly presented and well supported by visual diagrams.

# Weaknesses
- Although the paper claims that the proposed Trajectory-Centric Distillation effectively mitigates over-saturation and over-smoothing artifacts, the qualitative results do not convincingly demonstrate a clear superiority in visual quality. Specifically, it is unclear why Figure 4 does not use exactly the same text prompts and view settings as related works for direct comparison. In both Figure 4 and the Appendix, the results produced by Trajectory-Centric Distillation still exhibit noticeable smoothness and blurriness.

- The experimental setup appears questionable. While the main focus of the method is to improve text fidelity, the quantitative evaluation primarily relies on CLIP Score, which mainly reflects text-image alignment and does not adequately assess texture or geometric fidelity. Moreover, the visual results seem weaker than those in prior works, and the implementation details provided are insufficient, raising concerns about the fairness and reproducibility of the experiments.

- All experiments are conducted exclusively on Gaussian representations, and it remains unclear whether the proposed TraCe framework can outperform VSD when using NeRF representations.

---

> ### Author Rebuttal · Authors · 2025-07-31
>
> > **[W1] Qualitative Evaluation and Comparison**
>
> **On the Fairness of the Qualitative Comparisons:**
>
> The official code for some baseline methods we compare against—VSD, SDI, and CSD—are all implemented based on NeRF. To ensure a fair comparison, we utilized the 3DGS implementation within the Threestudio framework. Our parameter settings for the camera/pose are based on the settings for the 3DGS-based implementations from Threestudio. This is why our results may appear to differ from the NeRF-based view settings presented in their original papers.
>
> Furthermore, The text prompts used in our paper were sourced from the public DreamFusion online gallery [A]. These are standard, widely-used prompts that serve as a common benchmark in the community [20] [23] [29] [30] [35] [45] [48], ensuring our evaluation is relevant and consistent with established practices.
>
> ---
>
> **On Smoothness and Blurriness in Results:**
>
> The critique of smoothness often arises from comparisons to a specific family of methods, such as LucidDreamer (ISM) and Score Distillation via Inversion (SDI). These works aim to address over-smoothing by replacing the *stochastic random noising* of vanilla SDS with a deterministic term derived from **DDIM inversion**. This ensures the guidance is highly consistent, which is very effective for preserving sharp details. Our work, TraCe, introduces a novel and **orthogonal contribution**. We do not modify the stochastic noising process but instead provide **a more principled formulation for the new optimization objective** as a general version of SDS framework. **Therefore, these two categories of approaches are not mutually exclusive but are in fact complementary**. A very promising direction for future work is to combine our principled TraCe objective with a deterministic noising process from DDIM inversion.
>
> **Although TraCe was not designed to incorporate DDIM inversion specifically to solve the over-smoothing problem, it still effectively mitigates blurriness compared to prior methods.** For example, the intricate decorations on the cake (Figure 1), the sharp and defined patterns on the "hermit crab with a colorful shell" (Figure 4).
>
> ---
>
> [A] https://dreamfusion3d.github.io/gallery.html
>
> > **[W2] Experimental Rigor and Reproducibility**
>
> Regarding
> the explanation to experimental setup and more quantitative experiments, **please refer to the answer to Reviewer z2hB W1**.
>
> Regarding reproducibility, we have actually **provided the complete code for our project in supplementary materials** to ensure the reproducibility of our experiments.
>
> > **[W3] Generalization to NeRF**
>
> We use the official implementation of VSD. We keep other hyer-parameters the same for fair comparisons. We use prompts from DreamFusion gallery. TraCe outperforms VSD in both evaluation metrics.
>
> | **Method**              | **GPT-4o (GPTEval3D)-3D Plausibility** | **GPT-4o (GPTEval3D)-Texture Details** | **GPT-4o (GPTEval3D)-Geometry Details** | **GPT-4o (GPTEval3D)-Text-Asset Alignment** | **GPT-4o (GPTEval3D)-Text-Geometry Alignment** | **GPT-4o (GPTEval3D)-Overall** |
> | ----------------------- | -------------------------------------- | -------------------------------------- | --------------------------------------- | ------------------------------------------- | ---------------------------------------------- | ------------------------------ |
> | VSD (~5h)               | 1261.80                                | 1058.73                                | 1152.00                                 | 1246.37                                     | 1180.56                                        | 1012.50                        |
> | **TraCe (ours, ~3.5h)** | 1289.89                                | 1127.06                                | 1228.94                                 | 1284.60                                     | 1245.06                                        | 1179.86                        |
>
> | **Method**              | **ImageReward (T³Bench) $ \uparrow $** |
> | ----------------------- | ---------------------------- |
> | VSD (~5h)               | -0.6737 $\pm$ 0.4098            |
> | **TraCe (ours, ~3.5h)** | -0.4533 $\pm$ 0.4635            |
>
> > **[Q1] Fairness of Efficiency Comparison**
>
> Thank you for your comment.
>
> ---
>
> Regarding the optimization time for VSD, “the optimization time for ISM should be smaller than VSD” may be a misunderstanding. For ISM, we used its official 3DGS implementation with 5000 steps, which took approximately 20 minutes. For VSD, since our work uses a 3DGS representation, we experimentally determined its optimal settings for this different backbone. We found that performance peaked at around 2200 optimization steps. Training with fewer steps produced blurry assets, while training for longer introduced significant artifacts (e.g., large bright spots and widespread noisy colors), leading to degraded quality. We suspect this indicates that the VSD objective is less robust for 3DGS optimization. These 2200 steps yielded the best possible results for VSD in our setup and corresponded to the approximately 17-minute runtime reported in Table 1's quantitative comparisons in main paper.
>
> ---
>
> Regarding the optimization steps for our method, we note that while older methods required ~5000 steps, the number of training steps for recent 3DGS-based methods is considerably lower, typically ranging from 1200 (e.g., GaussianDreamer) to 2500 (e.g., DreamerXL) steps. Our choice of 1700 steps for TraCe was similarly determined by empirical convergence. Our experiments showed that generated assets were fully converged at this point, with no significant quality improvement in further steps. Therefore, this number was not chosen to artificially shorten the time but represents the point of optimal quality for our method.
>
> ---
>
> Finally, concerning the overhead of LoRA, our experiments show it adds only 1-2 minutes to the total generation time within our 3DGS framework. The significant quality improvements from TraCe are thus achieved with minimal additional computational cost.
>
> > **[Q2] Appropriateness of Evaluation Metrics**
>
> Please see our response to W2.
>
> > **[Q3] Comparison with Recent Works**
>
> Thank you for reviewer's suggestion. For our comparisons, we selected the five most commonly used and state-of-the-art methods from SDS-based works.
>
> In response to reviewer’s suggestion, we have now added new comparison experiments with two latest works. For details, please refer to the **tables in Reviewer XMDr W2**.
>
> > **[Q4] Ablation Study, Method Comparison and LoRA Analysis**
>
> Thank you for your suggestions. Regarding the ablation study, we have taken your feedback and conducted a new quantitative evaluation, which is detailed in the **table in Reviewer z2hB Q1**, and will be incorporated into the final version of the paper. We chose ImageReward, a community-recognized, SOTA evaluation method, as our evaluation metric.
>
> ---
>
> **In response to your question about the distinction between TraCe and the "Naive Bridge"**, the two approaches are fundamentally different. The Naive Bridge proposed in [30] does not rely on Equation (4). It constructs a bridge between distributions defined by two separate text prompts (i.e., a source prompt and a target prompt). Its guidance is based on the difference between the model's predicted scores conditioned on source prompt (e.g., negative prompts such as "oversaturated, smooth, pixelated...") and target prompt (the base prompt). This method steers the optimization by manipulating the text conditioning of a standard diffusion model to naively approximate a bridge.
>
> TraCe constructs its bridge between two images: the current rendering ($x_{rndr}$) and a dynamically estimated ideal target ($x_0^{pred}$). Our goal is to learn the score function of this specific image-to-image trajectory. Therefore, the guidance term in our Equation (4), subtracts the **analytically-defined ground-truth noise** that connects an intermediate point on the bridge ($x_t$) to the bridge's endpoint ($x_{rndr}$), which helps learn to accurately follow a direct transport trajectory between two distributions.
>
> Moreover, the Naive Bridge approach is not a standalone method and requires an initial training stage using standard SDS to be effective. TraCe, in contrast, is a complete, end-to-end framework that operates effectively from the start of optimization.
>
> ---
>
> **Regarding your questions about LoRA:**
>
> The assertion that LoRA "significantly increases optimization time" is actually not the case in our framework. Our experiments show that incorporating LoRA adds **only 1-2 minutes** to the total generation time per asset.
>
> This minimal cost is well-justified by the performance gains, as LoRA is crucial for addressing a key distributional mismatch. A standard diffusion model was not trained to predict the score for a Schrödinger Bridge between two arbitrary distributions (our current rendering and our ideal target). Applying it directly to this new task leads to prediction errors, which manifest as the artifacts you noted (e.g., incorrect colors, blurriness). Therefore, TraCe uses LoRA to specialize the diffusion model for this exact task. By training the LoRA parameters on samples along the bridge, we explicitly teach the model the score dynamics of this optimal, direct trajectory.
>
> Additionally, the role of LoRA is not just limited to color correction. Our experiments show that LoRA also improves the details and sharpness of the generated assets and eliminates the formation of bright spots. Examples include the texture of animal fur, the details and clarity of flowers, and the intricacy of patterns on an axe, among others. Our new quantitative ablation **in table in Reviewer z2hB Q1** confirms this, showing that adding LoRA yields a more significant performance improvement than only adding scheduled t-sampling (ImageReward score from -0.3389 to -0.2486). This demonstrates its role is critical, not peripheral.

---

> > ### Comment · Reviewer_ifTq · 2025-08-06
> >
> > Thank you for clarifying the computational overhead of LoRA; that was helpful. However, I still have several concerns that require your attention:
> > 1. Regarding the Validity of the T$^3$Bench Experiments:
> > How did you obtain negative scores using the T$^3$Bench evaluation? Based on my understanding of its evaluation code, I don't see how negative results would be produced. Please clarify your evaluation protocol.
> > Camera Parameters: I disagree with your decision to use unified camera parameters for all reproduced methods. In my experience, these settings may impact the final results. Adhering to each method's original setup is the most appropriate way to ensure a fair comparison, as your settings may disadvantage other methods.
> >
> >
> > 2. Regarding the Credibility of Results and Visualization Quality:
> > I've noticed from your provided config files that your method does not seem to employ common techniques like DreamTime or annealing schedules. This is likely a reason for the poor quality of the visualizations presented in your paper.
> > Given the quality of your current visualizations, I find it difficult to believe that your method quantitatively outperforms DreamerXL on image quality metrics. I need to discuss this with other reviewers about the quality of the visualization due to the motivation of this paper is improving texture quality. While you claim in the paper that your method is orthogonal to techniques like ISM and TSM, this claim is not substantiated by experimental evidence and cannot be accepted by assumption alone. Furthermore, upon examining the results in Appendix J, I do not observe a significant quality improvement over your baseline GaussianDreamer.

---

> > > ### Author Response · Authors · 2025-08-07
> > > **Response**
> > >
> > > We thank the reviewer for the detailed feedback.
> > >
> > > ---
> > >
> > > We would like to clarify that for the T³Bench evaluation, we reported the **raw ImageReward** scores, which the benchmark is based on. The ImageReward model produces scores that are naturally centered around zero, so negative values are an expected and standard outcome for many generations. T³Bench's final reported score is simply a scaled version (0-100) of this raw value; we provided the original for transparency.
> > >
> > > ---
> > >
> > > The reviewer is correct that the original camera hyper-parameters for VSD, CSD, and SDI were carefully tuned. However, they were tuned for NeRF, and forcing NeRF-tuned parameters onto a 3DGS backend would lead to a critical representation-data mismatch. Actually **the only change compared with original camera hyper-parameters for VSD, CSD, and SDI is camera distance range**. VSD, SDI and CSD all use a narrower camera distance range ([1.5,2.0]), and we use [1.5,4.0]. **However, through our experiments, we found that 3DGS models benefit from a wider range of camera distance to learn a more robust 3D shape (with IR score higher about 0.5)**. The new camera distance range were the result of empirical testing to find a configuration that allowed **all distillation algorithms (VSD, CSD, SDI, and our TraCe) to produce their best possible results on the 3DGS, while not to disadvantage their performances.** So, we borrow the this hyper-parameter for the 3DGS-based implementations from Threestudio, and it is tuned for all baselines, not for our method.
> > >
> > > ---
> > >
> > > We would like to clarify that our method *does* employ an annealing schedule, which is the term “Scheduled t-sampling” in our paper. We made a change to original annealing strategy: we let the t for x_t (which is the sampled point along the Schrödinger Bridge trajectory) to be annealed to learn the whole trajectory starting from the source rendering to the target image. Our experiments show that this change can achieve better results.
> > >
> > > ---
> > >
> > > While DreamerXL can achieve superior results in some cases (e.g., “zombie joker”), this is expected given its stronger base model. DreamerXL uses the more powerful SDXL as the base model, which is trained on highly aesthetic images; in our initial submission, all our experiments used SDv2.1 as the base model because our goal was to have fair comparison to other score distillation methods (including VSD, SDI, etc.) that are also based on SDv2.1. It is worth noting, however, that even with a weaker base model, TraCe's principled optimization allows it to achieve better on other prompts like “a plate piled high with chocolate chip cookies”, "a highly-detailed sandcastle", and “a pineapple”, etc.
> > >
> > > ---
> > >
> > > We respectfully disagree with the assessment that there is no significant quality improvement over GaussianDreamer and would like to guide the reviewer's attention to these specific improvements:
> > >
> > > 1. Superior Geometric Coherence and Detail: "A blue lobster" and “lady bug”: Please compare the legs of lobster and claws of lady bug. The GaussianDreamer appendages and legs are noisy and muddled, lacking clear form. In contrast, our result shows distinct, well-structured claws and legs with more realistic anatomy.
> > > 2. Enhanced Textural Fidelity and Photorealism: "A candelabra...": Note the flames and the material of the candelabra itself. Our method produces crisp, distinct flames, while the baseline's are blurry halos. Furthermore, our candelabra has a more defined metallic texture, whereas the baseline appears softer and less detailed.
> > > 3. Improved Prompt Adherence and Stylistic Control: "A crab, low poly": This prompt requests a specific artistic style. The GaussianDreamer result is noisy, and the "low poly" nature is not clearly defined. Our result more successfully captures the aesthetic, with cleaner, more distinct polygonal facets and sharper edges, demonstrating superior adherence to the user's prompt. For "A golden goblet, low poly": This trend is also visible as seen in Figure 4 of the main paper. The result from GaussianDreamer is blurry and fails to capture the "low poly" style, rendering a smooth object with a dull, plastic-like texture. In contrast, our result clearly exhibits the requested "low poly" faceted geometry and a convincing metallic gold finish, demonstrating both superior stylistic interpretation and material realism.

---

> > > > ### Comment · Reviewer_ifTq · 2025-08-07
> > > >
> > > > Thanks for the response which solve my concern about fair comparison. I will raise score and authors should claim the baseline settings and provide NeRF-based quantity and quality comparison in the revision.

---

> > > > > ### Author Response · Authors · 2025-08-08
> > > > > **Thanks for your response**
> > > > >
> > > > > Dear Reviewer ifTq,
> > > > >
> > > > > We sincerely thank you for your insightful review and encouraging feedback on our manuscript. Your comments have been incredibly helpful and have provided us with a clear path for strengthening the paper. In the revised paper, we will make modifications based on the recommendations provided by all reviewers and claim the baseline settings and provide NeRF-based quantity and quality comparison.
> > > > >
> > > > > We are confident that these revisions will significantly improve the quality and clarity of our work. Thank you once again for your time, expertise, and constructive assessment.
> > > > >
> > > > > Sincerely,
> > > > >
> > > > > The Authors of paper ID 4750.

---

> ### Author Response · Authors · 2025-08-05
> **Reminder**
>
> Dear Reviewer ifTq,
>
> Thank you for taking the time to review our paper. We appreciate your efforts in helping us improve our work.
>
> As we approach the end of the discussion period, may we ask you to kindly check the rebuttal? Many thanks for your kind attention.
>
> Thank you.
>
> The Authors of paper ID 4750.

---

> ### Author Response · Authors · 2025-08-08
> **Thank you**
>
> Dear Reviewer ifTq,
>
> Thank you so much for raising the score for our manuscript! More importantly, thank you so much for your positive feedback on our rebuttal and for confirming that our responses addressed your major concerns. We were very grateful and encouraged to read your message.
>
> Thank you again for your time and invaluable support for our work. We truly appreciate it.
>
> Sincerely,
>
> The Authors of Paper ID 4750

---

### Official Review · Reviewer_iYfs · 2025-07-01

**Clarity:** 3
**Significance:** 3
**Originality:** 3
**Rating:** 4
**Confidence:** 3

**Summary:**

Summary:

The paper theoretically establishes Score Distillation Sampling (SDS) as a simplified instance of the Schrödinger Bridge and introduces the TraCe framework, which enables more stable 3D optimization. Experiments demonstrate that TraCe outperforms existing methods in terms of visual quality and text alignment.

**Questions:**

1、Please clarify the novelty of the proposed work.

2、Please clarify the differences between this work and prior works with similar ideas, such as DreamFlow [2] and I²SB [1].

3、Why the generated results still have artifacts such as over-saturation and over-smoothing?

[1] Liu, Guan-Horng, et al. "I $^ 2$ SB: Image-to-Image Schr\" odinger Bridge." arXiv preprint arXiv:2302.05872 (2023).

[2]Lee, Kyungmin, Kihyuk Sohn, and Jinwoo Shin. "Dreamflow: High-quality text-to-3d generation by approximating probability flow." arXiv preprint arXiv:2403.14966 (2024).

**Ethical Concerns:**

["NO or VERY MINOR ethics concerns only"]

**Final Justification:**

I have considered rebuttal and discussions with authors, other reviewers and AC. I raise my score to 4 (borderline accept).

**Limitations:**

yes

**Paper Formatting Concerns:**

No major formatting issues.

**Quality:**

2

**Strengths And Weaknesses:**

Strengths:

This paper presents quantitative and qualitative comparisons with other methods.

Weaknesses：

1、A related work has proved that Score-based Generative Model (SGM) is a special case of Schrödinger Bridge [1].

2、This work adopts the tractable Schrödinger Bridge (SB) framework introduced in related work.

3、The results of the proposed method have still artifacts such as over-saturation and over-smoothing.

4、The paper lacks a comparison with Dreamflow[2].

5、There are formatting issues in the references, e.g., Reference 26.

[1] Liu, Guan-Horng, et al. "I $^ 2$ SB: Image-to-Image Schr\" odinger Bridge." arXiv preprint arXiv:2302.05872 (2023).

[2]Lee, Kyungmin, Kihyuk Sohn, and Jinwoo Shin. "Dreamflow: High-quality text-to-3d generation by approximating probability flow." arXiv preprint arXiv:2403.14966 (2024).

---

> ### Author Rebuttal · Authors · 2025-07-31
>
> We thank the reviewer for your time and feedback. However, we believe there have been several fundamental misunderstandings regarding our paper's novelty, contributions, and results. We would like to take this opportunity to clarify these points.
>
> > **1. On the Novelty of Our Work (Q1 & Q2; W1, W2, W4)**
>
> The reviewer’s primary concerns appear to stem from a misunderstanding of our work's core novelty.
>
> **Our Theoretical Contribution (vs. W1):** We would like to clarify a key distinction. Our paper does **not** claim that "SGM is a special case of Schrödinger Bridge" is a novel finding. We state this as established background in our preliminaries (Section 3). Our novel theoretical contribution, as stated in the introduction and proven in Section 4.1, is to be the **first to formally establish that Score Distillation Sampling (SDS) is a simplified instance of the Schrödinger Bridge framework**. This insight allows us to formulate TraCe to learn a direct, optimal transport trajectory to the target, which enables the generation of higher quality and fidelity assets with lower and more stable CFG values.
>
> **Our Methodological Contribution (vs. W2 & I2SB [26]):**  Building upon the Schrödinger Bridge (SB) framework is standard practice in community, such as [A] [B] [C]. **Our novelty lies in how to better solve the problem of score distillation for text-to-3D generation with the use of this framework.** Our TraCe introduces a new formulation where we construct a diffusion bridge specifically from the current 3D render ($x_{rndr}$) to its dynamically estimated denoised target ($x_0^{pred}$). We then train a LoRA-adapted model to learn the score dynamics of this specific trajectory. This unique formulation is the core of TraCe and is not presented in I2SB [26] or other prior works.
>
> **Comparison with DreamFlow [20] (vs. W4):** We explicitly discussed DreamFlow and differentiate our work in Section 2 (Related Work) of the submission. As we note, DreamFlow "proposes to approximate the backward Schrödinger Bridge dynamics between rendered and target views by simply repurposing a fine-tuned text-to-image model, a heuristic potentially deviating from the true underlying Schrödinger Bridge process". It is **a heuristic approach that approximates the Schrödinger Bridge dynamics, unfortunately without the theoretical analysis that underpins our work**. In contrast, TraCe is more principled: we explicitly construct the Schrödinger Bridge and train a model on its true score dynamics grounded in the analytical formulation, leading to a more stable and direct optimization trajectory.
>
> > **2. On the Quality of Generated Results (Q3; W3)**
>
> We are sorry some misunderstanding is caused to you. In fact, a primary motivation and demonstrated contribution of our work is the **significant reduction** of these very artifacts compared to prior works.
>
> **Superior Fidelity:** Throughout the qualitative comparisons in Figure 4, our method consistently produces sharper details and more naturalistic textures (e.g., the fur on the "fox," the details of the "wedding cake", the texture of “golden globe”) compared to strong baselines including VSD, SDS, CSD, ISM, etc.
>
> **Direct Visual Evidence:** In Figure 1, the results from standard SDS and VSD show clear artifacts (e.g., over-saturation, blurriness), while our result is visibly cleaner and more detailed at a fair CFG value.
>
> **Qualitative and Quantitative Results:** This visual superiority is backed by our state-of-the-art quantitative CLIP scores reported in Table 1: Quantitative comparisons. We provide qualitative comparisons in the main paper (Figures 4) and supplementary materials (Figures 4, 7-12) that visually demonstrate our method's superiority. More importantly, we conducted a comprehensive user study with 60 participants (Supplementary Section D) showing a clear user preference for TraCe, and we believe human perceptual evaluation is often more persuasive than automated metrics [20]. We also provide additional quantitative experiments, and please refer to the **tables in Reviewer XMDr W2**.
>
> > **3. Minor Formatting Issues (W5)**
>
> Thank you! We have corrected such issues and will ensure the paper will be thoroughly proofread in the revised version.
>
> ---
>
> We hope these clarifications address your concerns, and effectively communicate the novelty of our theoretical insights, our proposed TraCe framework, and the significance of our experimental results. Thank you again.
>
> ---
>
> [A] Xuan Su, Jiaming Song, Chenlin Meng, and Stefano Ermon.
> Dual diffusion implicit bridges for image-to-image translation. In International Conference on Learning Representations, 2023.
>
> [B] Yuyang Shi, Valentin De Bortoli, Andrew Campbell, and Arnaud Doucet. Diffusion schrödinger bridge matching. Neural Information Processing Systems (NeurIPS), 2023.
>
> [C] S. Diefenbacher, G.-H. Liu, V. Mikuni, B. Nachman, and W. Nie, Improving generative model-based unfolding with Schrödinger bridges, Phys. Rev. D 109 (2024), no. 7 076011, [arXiv:2308.12351].

---

> > ### Comment · Area_Chair_dmgH · 2025-08-06
> > **Please_respond**
> >
> > Hello, Would greatly appreciate your response to the authors' rebuttal particularly at this critical point in time. Thanks
> > Best

---

> > ### Comment · Reviewer_iYfs · 2025-08-08
> >
> > Thank you for the rebuttal. My major concerns have been addressed. I will raise score.

---

> ### Comment · Area_Chair_dmgH · 2025-08-04
> **Rebuttal_of_Paper4750**
>
> Dear Reviewer  iYfs ,
> I would greatly appreciate your prompt evaluation of the authors' rebuttal to some of your comments, particularly concerning the
> - The SDS and SGM equivalence in their relation to the S. Bridge.
> - The authors' failure to consider   Dreamflow and acknowledge its comparative performance.
> - Other comments responded to in the rebuttal.
> Thank you.
> Best regards,

---

> ### Author Response · Authors · 2025-08-05
> **Reminder**
>
> Dear Reviewer iYfs,
>
> Thank you for taking the time to review our paper. We appreciate your efforts in helping us improve our work.
>
> As we approach the end of the discussion period, may we ask you to kindly check the rebuttal? Many thanks for your kind attention.
>
> Thank you.
>
> The Authors of paper ID 4750.

---

> ### Author Response · Authors · 2025-08-08
> **Thank you**
>
> Dear Reviewer iYfs,
>
> Thank you so much for raising the score for our manuscript! More importantly, thank you so much for your positive feedback on our rebuttal and for confirming that our responses addressed your major concerns. We were very grateful and encouraged to read your message.
>
> Thank you again for your time and invaluable support for our work. We truly appreciate it.
>
> Sincerely,
>
> The Authors of Paper ID 4750

---

> ### Author Response · Authors · 2025-08-09
> **Gentle Reminder**
>
> Dear Reviewer iYfs,
>
> We are writing to send a gentle reminder about the rating update and mandatory acknowledgement deadline for reviews. I understand this final step includes submitting the finalized rating for the paper.
>
> We know this is an incredibly busy time and sincerely appreciate all the effort and time you have already dedicated to reviewing our work.
>
> Thank you for your consideration.
>
> Sincerely,
>
> The Authors of Paper ID 4750

---

### Official Review · Reviewer_XMDr · 2025-07-03

**Clarity:** 3
**Significance:** 3
**Originality:** 3
**Rating:** 4
**Confidence:** 4

**Summary:**

This paper tackles the problem of artifacts such as over-saturation and over-smoothing in SDS optimization-based text-to-3D generation methods by theoretically connecting SDS to the Schrödinger Bridge framework and showing that SDS is a simplified instance of this optimal transport approach. The authors introduce Trajectory-Centric Distillation (TraCe), which explicitly constructs a diffusion bridge between current renderings and their text-conditioned targets, training a LoRA-adapted model on the trajectory's score dynamics to enable high-quality generation with smaller Classifier-free Guidance values.

**Questions:**

Please see the weaknesses section.

**Ethical Concerns:**

["NO or VERY MINOR ethics concerns only"]

**Final Justification:**

After careful consideration of the rebuttal and discussions, I raise my recommendation toward acceptance based on the following assessment:

- The theoretical contribution interpreting SDS through the Schrödinger Bridge perspective is novel, robust, and significant, providing meaningful advancement to score distillation-based 2D/3D generation.
- The mathematical framework and proofs supporting this interpretation are sound and well-established.
- The authors successfully translate this theoretical insight into a practical methodology, effectively using rendered image and one-step denoised image as Schrödinger Bridge endpoints.
- During the rebuttal phase, the authors have provided extensive results to convince me of their method's effectiveness, even though shortcomings remain. Still, I see this paper has meaningful contributions to the field which can be expanded upon by later work.

**Limitations:**

Yes.

**Quality:**

2

**Strengths And Weaknesses:**

**Strengths:**
- The paper is well-written, highly organized and easy to follow.
- The contribution of the paper in interpreting SDS in the perspective of Schrödinger Bridge is novel, robust and significant, providing meaningful contribution to the field of score distillation based 2D / 3D generation. Theoretical explanation and proof of this interpretation, provided in the paper is robust and sound.
- The authors translate this interpretation to the methodology well, using the rendered image $x_{rndr}$ and one-step denoised image $x_{0}^{pred}$ as two endpoints of the Schrödinger Bridge and formulating the optimization process as interpolation of two diffusion denoising paths.

**Weaknesses:**
- The improvement from this method, shown in the experimental section of the paper, seems to be somewhat marginal, unable to achieve meaningful modification in overall geometry but only in texture. Can the authors provide examples in which i) baseline result produces divergent 3D scene (such as Janus problem) but TraCe results in significant improvements from it, or ii) results in which generated result's geometry and layout differs significantly due to usage of TraCe? This seems to be my main concern with the paper, and if this point is elaborated (regarding specifically *how* TraCe improves upon SDS process in implementation), I would raise my score toward acceptance.
- Both quantitative and qualitative results seem to be lacking in the main paper: can additional experiments (especially qualitative) be provided to back up that TraCe is effective in practice and not only in theory?
- The paper is unclear in how the LoRA-adapted model $\epsilon_\phi(x_t, t, y, c)$ is trained to predict the noise from $x_t$ to $x_{0}^{pred}$: please elaborate. Is it trained online during optimization like LoRA-adapted model in ProlificDreamer (that does seem to be likely)? If this is the case, can this method be seen as a variant of VSD under single-particle setting?

---

> ### Author Rebuttal · Authors · 2025-07-31
>
> > **[W1] Addressing Geometric Improvements and the Janus Problem**
>
> Thanks for the comment. The reviewer’s main concern is understanding *how* our TraCe improves upon the SDS process to justify its results, particularly concerning geometry.
>
> **The primary goal of TraCe is not to solve fundamental 3D-awareness issues like the Janus problem or to generate entirely different scene layouts from baselines.** Instead, TraCe targets a different, but equally critical, challenge in text-to-3D generation: **the quality and the fidelity of the optimized 3D asset**. Like other works [20] [23] [29] [30] [45] [48] that aim to refine the distillation process, our focus is on improving the quality and fidelity of the final result by addressing instabilities and artifacts inherent in the standard SDS framework.
>
> The proposed TraCe makes significant and consistent refinements to geometric quality and coherence. Our qualitative results provide several examples of this:
> 1. TraCe is highly effective at removing geometric artifacts. For instance, in "a candelabra with many candles on a red velvet tablecloth" (Figure 7 in supplementary), it replaces the outrageous flame geometry from GaussianDreamer with controlled, plausible flames.
> 2. TraCe generates finer structural details. When generating "A gothic cathedral with stained glass windows and tall arches" (Figure 8 in supplementary), the cross atop the main spire has a clear and reasonable geometry, a detail that is misshapen in the ProlificDreamer output.
> 3. For the "blue lobster" and "a ladybug" (Figure 7 in supplementary), our method eliminates the extraneous geometry seen in the baseline, such as the lobster's massive, disjointed green antennae and the ladybug's misshapen feet.
> 4. In "a mug of hot chocolate with whipped cream and marshmallows" (Figure 10 in supplementary), the distracting "cloud" of color from Classifier Score Distillation is gone, resulting in a more coherent shape.
>
> ---
>
> Regarding the Janus (multi-face) problem, we agree this is a crucial challenge. However, this is a known limitation stemming from the underlying 2D text-to-image prior, which inherently lack 3D awareness. Therefore, like other SDS-based methods [20] [23] [29] [30] [45] [48], etc., TraCe is not designed to be a fundamental solution to this issue. We use standard mitigation techniques like negative prompting to mitigate the Janus problem. One way to address Janus problem is to use 3D-aware multi-view diffusion model (e.g., MVDream), and our framework is compatible with such models for 3D optimization. This will be discussed in revised version.
>
> > **[W2] Additional Qualitative and Quantitative Results**
>
> We thank the reviewer for their valuable feedback on the presentation of our results.
>
> ---
>
> For our quantitative evaluation, we used the **CLIP score** and **a detailed user study**. This multi-faceted approach was chosen deliberately. The CLIP score aligns our work with the evaluation practices of prominent text-to-3D papers like ProlificDreamer [45], DreamFlow [20], and Score Distillation via Reparametrized DDIM [29]. At the same time, human perceptual evaluation is often more persuasive than automated metrics [20]. Therefore, following the example of methods like DreamFlow, we conducted a rigorous user study with 60 participants referring to DreamFlow’s user study settings, which shows a consistent preference for TraCe in 3D consistency, prompt fidelity, and photorealism.
>
> In response to reviewer’s suggestion, we conduct additional quantitative experiments, the results of which are available in the tables below. We selected two community-recognized and common evaluation metrics for the task of 3D asset generation: GPT-4o (with an implementation based on GPTEval3D) and ImageReward (with an implementation based on T³Bench). The combination of the user study, CLIP scores, and the extensive qualitative results in the supplementary material provides strong practical backing for the theoretical advancements presented in our work.
>
> ---
>
> Regarding the qualitative results, we presented key comparisons in the main paper due to page limitations, with the full, extensive evidence contained in the supplementary material. We respectfully refer the reviewer to the supplementary document, which provides a comprehensive demonstration of TraCe's practical effectiveness across numerous examples and against six state-of-the-art methods (Figure 4 in main manuscript and Figures 7-12 in supplementary). The supplementary also includes detailed gradient visualizations (Figure 4 in supplementary) , an ablation study of the CFG value (Figure 1 in supplementary)). We will provide more in supplementary in revision.
>
> | Method                        | GPT-4o (GPTEval3D) - 3D Plausibility | GPT-4o (GPTEval3D) - Texture Details | GPT-4o (GPTEval3D) - Geometry Details | GPT-4o (GPTEval3D) - Text-Asset Alignment | GPT-4o (GPTEval3D) - Text-Geometry Alignment | GPT-4o (GPTEval3D) - Overall |
> | ----------------------------- | ------------------------------------ | ------------------------------------ | ------------------------------------- | ----------------------------------------- | -------------------------------------------- | ---------------------------- |
> | **sds** (11min)               | 953.43                              | 1022.40                             | 1003.04                              | 1038.08                                  | 1023.93                                     | 1018.09                     |
> | **vsd** (17min)               | 1020.81                             | 985.59                              | 996.29                               | 993.04                                   | 1006.36                                     | 1007.49                     |
> | **csd** (11min)               | 1009.29                             | 953.43                              | 984.24                               | 1024.24                                  | 950.82                                      | 983.04                      |
> | **sdi** (10min)               | 1002.51                             | 949.00                              | 970.09                               | 981.04                                   | 973.58                                      | 971.98                      |
> | **ism** (20min)               | 1003.79                             | 951.63                              | 998.94                               | 1007.07                                  | 989.78                                      | 1012.37                     |
> | **consistency** (new, 14 min) | 981.48                              | 1001.88                             | 1013.99                              | 986.38                                   | 995.42                                      | 993.79                      |
> | **DreamerXL** (new, ~1h)        | 1005.58                             | 997.22                              | 1002.94                              | 1032.20                                  | 1012.37                                      | 1014.49                     |
> | **TraCe** (ours, 14min)       | 1025.42                             | 1031.98                             | 1014.03                              | 1032.41                                  | 1024.45                                     | 1028.03                     |
>
> | **Method**               | **ImageReward (T³Bench) $ \uparrow $** |
> | ------------------------ | ---------------------------- |
> | SDS (11min)              | -0.4329 $ \pm $ 0.9125              |
> | vsd (17min)              | -0.5330 $ \pm $ 0.8927              |
> | csd (11min)              | -0.6715 $ \pm $ 0.7482              |
> | sdi (10min)              | -0.8334 $ \pm $ 1.0391              |
> | ism (20min)              | -0.3904 $ \pm $ 0.9503              |
> | consistency (new, 14min) | -0.5497 $ \pm $ 0.9200            |
> | DreamerXL (new, ~1h)       | -0.3382 $ \pm $ 0.9225           |
> | TraCe (ours, 14min)      | -0.2855 $ \pm $ 0.8909           |
>
> > **[W3] Methodological Details and Comparison to VSD**
>
> We thank the reviewer for this insightful question and the opportunity to clarify our method. You are correct that the LoRA-adapted model is trained online, concurrently with the 3D model. However, the underlying principle of our TraCe framework is fundamentally different from that of Variational Score Distillation (VSD).
>
> The key differences are as follows:
>
> In VSD, the LoRA-adapted model is trained to approximate the score of the entire distribution of current (source) renderings, which is also mentioned in [30]. Its goal is to perform variational inference by matching two distributions.
>
> In contrast, **our TraCe framework does not model the distribution of currrent** **renderings**. This is where the novelty of TraCe's LoRA training lies. Instead of learning a distributional score like VSD, our LoRA model $\epsilon_{\phi}$ is trained via a distinct objective: **learning the transition of the point x_t on the bridge trajectory between the current rendering $x_{rndr}$ and the ideal target $x_0^{pred}$.** This provides direct supervision for the score function along this specific trajectory. This online adaptation is essential, as a pre-trained T2I model has no knowledge of the dynamics of our constructed Schrödinger Bridge between two arbitrary distributions. We are not merely specializing the model, but rather training it on a fundamentally different and more direct learning signal than the distributional objective used in VSD.
>
> Also, we appreciate the reviewer pointing out this potential ambiguity and will revise Section 4.2 in revision to make this critical distinction more explicit.

---

> ### Author Response · Authors · 2025-08-05
> **Reminder**
>
> Dear Reviewer XMDr,
>
> Thank you for taking the time to review our paper. We appreciate your efforts in helping us improve our work.
>
> As we approach the end of the discussion period, may we ask you to kindly check the rebuttal? Many thanks for your kind attention.
>
> Thank you.
>
> The Authors of paper ID 4750.

---

> > ### Comment · Area_Chair_dmgH · 2025-08-06
> > **Response_to_authors'Rebuttal**
> >
> > Hello,
> > Would greatly appreciate your response to the authors' rebuttal.
> > Thanks

---

> ### Comment · Reviewer_XMDr · 2025-08-07
>
> I thank the authors for the detailed rebuttal.
>
> However, I have further concerns: the authors assert that their main objective is improving quality and fidelity of optimized 3D assets, yet the performance differences are marginal in both qualitative results (paper) and quantitative results (paper and rebuttal).
>
> I particularly question the reliability of the new quantitative results provided. For example, the texture-detail metric shows VSD performing far worse than SDS, which contradicts established knowledge—VSD typically brings dramatic improvements over SDS, especially in texture quality and fidelity. Conversely, while VSD is generally more prone to Janus problems than SDS, the "3D plausibility" metric shows VSD scoring far higher than SDS. These results contradict well-established field knowledge, raising concerns about the evaluation process reliability.
>
> I believe this discrepancy stems from the authors' exclusive use of 3DGS as their 3D representation backbone. In my experience, score distillation sampling behaves very differently with implicit versus explicit 3D representations. For instance, VSD and ISD's [1] effectiveness is dramatically different between Instant-NGP and 3DGS representations—far more effective with Instant-NGP and less effective with 3DGS. Similarly, CFG scale variations impact Instant-NGP baselines much more severely than 3DGS baselines. Therefore, the proposed diffusion process's effectiveness would be more clearly demonstrated with Instant-NGP/NeRF baselines.
>
> **I ask the authors to conduct experiments on Instant-NGP baselines to verify this method's effectiveness.** While time is limited and I apologize for this late request (I will not lower my score further, and extensive results are not required), I want to verify whether this method has impact as dramatic as VSD's improvement over naive SDS. If clearly verified, I will consider raising my score. However, the authors have chosen a confusing experimental setting that at best reduces their method's apparent impact and at worst conflicts with established field knowledge, making evaluation results dubious and potentially confusing for readers.
>
> [1] LucidDreamer: Towards High-Fidelity Text-to-3D Generation via Interval Score Matching, Liang et al., CVPR 2024

---

> ### Author Response · Authors · 2025-08-08
> **Response Part1**
>
> > **On the Reliability of Quantitative Results and Backbone Choice**
>
> We thank the reviewer for their insightful analysis of our new quantitative results. Their observation that VSD's performance on a 3DGS backbone contradicts its well-established, superior behavior on NeRF/Instant-NGP is ***entirely correct and highlights the core motivation for our experimental design***.
>
> As the reviewer astutely noted, score distillation methods behave very differently on implicit vs. explicit representations. **Our own experiments confirm this: on 3DGS, we found VSD’s texture and geometry improvements over SDS were not as pronounced as they are on NeRF.** This discrepancy is precisely why the automated GPTEval3D metric, when applied to 3DGS-based results, produced the counter-intuitive scores the reviewer pointed out. (We provide more discussion in the below paragraph labeled with “*”.)
>
> This brings us to why we chose a 3DGS-only setting for our main comparison. In the 3D computer vision community, 3DGS is rapidly superseding NeRF-based methods due to its superior rendering quality and efficiency, making it the more relevant and forward-looking representation for text-to-3D research. The very fact that an established method like VSD does not seamlessly transfer its benefits to 3DGS highlights a gap in the field—a need for a distillation technique that is robust and effective specifically for explicit representations. TraCe was designed to fill this gap. Furthermore, as the reviewer notes, score distillation methods behave very differently on implicit versus explicit representations. Therefore, **proving that TraCe works effectively on the challenging 3DGS backbone is, in itself, a robust validation of our method's contribution**.
>
> *We would also like to explain further about the Texture and 3D Plausibility evaluation between SDS and VSD with 3DGS: (1) Texture: Through our experiments, the texture of VSD outputs is worse than SDS. Figure 1 is a clear illustration, and we invite the reviewer to compare the standard VSD in column a with the standard SDS in column b. For VSD, the results are notably blurry. The cake's sprinkles are indistinct, and the frosting lacks sharp definition. The goblet's surface is murky and devoid of fine detail. In contrast, the SDS results are significantly sharper and have more details. The frosting on the cake is more clearly defined, and the goblet has a more convincing metallic texture.(2) 3D Plausibility: The Janus problem is a well-known failure mode for VSD. However, *3D plausibility evaluation in GPTEval3D is multifaceted*. Standard SDS, particularly with 3DGS, is prone to different artifacts, such as floating geometry, disconnected components, and hollow, "facade-like" structures. We hypothesize that the GPTEval3D plausibility metric is more heavily penalized by these signs of volumetric incompleteness than by the semantic error of a multi-faced object.

---

> ### Author Response · Authors · 2025-08-08
> **Response Part2**
>
> > **On Demonstrating a "Dramatic Improvement" over SDS**
>
> Regarding the reviewer's concern about verifying a dramatic improvement over the naive SDS baseline, we respectfully disagree with the assessment that there is no significant quality improvement over SDS and would like to guide the reviewer's attention to these specific improvements:
>
> **Overcoming Core SDS Failure Modes at Fair Guidance:** Perhaps the clearest illustration of TraCe's advantage is the direct comparison with standard SDS at a fair and stable CFG value, as shown in Figure 1 (main paper). This comparison highlights how our method mitigates common SDS artifacts and achieves higher quality and fidelity.
>
> Furthermore, Figure 3 (Supplementary) shows a direct **2D** image optimization where TraCe produces a significantly clearer result than the lowly-detailed, over-saturated outputs from SDS. The 2D experiment is designed as a diagnostic tool to isolate and directly compare the behavior of different score distillation gradients, removing the complexities of 3D representations. This type of analysis, where a 2D image is optimized directly, has been used in prior work (e.g., VSD, etc) to understand the inherent properties of the guidance signal itself. This analysis provides evidence that our TraCe framework offers a fundamentally more stable and accurate guidance signal. This improvement in the 2D domain directly translates to the superior quality we observe in the more complex task of text-to-3D generation.
>
> More importantly, Figure 4 (main paper), Figure 7 (Supplementary) presents **3D** comparisons exhibit a dramatic leap over the artifact-ridden SDS baseline:
>
> 1. Superior Geometric Coherence and Detail: *"A blue lobster"* and *“lady bug”* (Figure 7 (Supplementary)): Please compare the legs of lobster and claws of lady bug. The GaussianDreamer appendages and legs are noisy and muddled, lacking clear form. In contrast, our result shows distinct, well-structured claws and legs with more realistic anatomy.
>
> 2. Enhanced Textural Fidelity and Photorealism: *"A candelabra..."* (Figure 7 (Supplementary)): Note the flames and the material of the candelabra itself. Our method produces crisp, distinct flames, while the baseline's are blurry halos. Furthermore, our candelabra has a more defined metallic texture, whereas the baseline appears softer and less detailed.
>
> 3. Improved Prompt Adherence and Stylistic Control: *"A crab, low poly"* (Figure 4 (main paper))**:** This prompt requests a specific artistic style. The GaussianDreamer result is noisy, and the "low poly" nature is not clearly defined. Our result more successfully captures the aesthetic, with cleaner, more distinct polygonal facets and sharper edges, demonstrating superior adherence to the user's prompt. For *"A golden goblet, low poly"* (Figure 4 (main paper))**:** This trend is also visible as seen in Figure 4 of the main paper. The result from GaussianDreamer is blurry and fails to capture the "low poly" style, rendering a smooth object with a dull, plastic-like texture. In contrast, our result clearly exhibits the requested "low poly" faceted geometry and a convincing metallic gold finish, demonstrating both superior stylistic interpretation and material realism.

---

> ### Author Response · Authors · 2025-08-08
> **Response Part3**
>
> > **Additional verification experiment**
>
> We also provide an additional verification experiment below for proving TraCe’s generalization to NeRF. We use the official implementation of SDS and VSD, and we use prompts from DreamFusion online gallery for this experiment.
>
> | **Method**              | **GPT-4o (GPTEval3D)-3D Plausibility** | **GPT-4o (GPTEval3D)-Texture Details** | **GPT-4o (GPTEval3D)-Geometry Details** | **GPT-4o (GPTEval3D)-Text-Asset Alignment** | **GPT-4o (GPTEval3D)-Text-Geometry Alignment** | **GPT-4o (GPTEval3D)-Overall** |
> | ----------------------- | -------------------------------------- | -------------------------------------- | --------------------------------------- | ------------------------------------------- | ---------------------------------------------- | ------------------------------ |
> | SDS (~1h)               | 1000.00                                | 1000.00                                | 1000.00                                 | 1000.00                                     | 1000.00                                        | 1000.00                        |
> | VSD (~5h)               | 1261.80                                | 1058.73                                | 1152.00                                 | 1246.37                                     | 1180.56                                        | 1012.50                        |
> | **TraCe (ours, ~3.5h)** | 1289.89                                | 1127.06                                | 1228.94                                 | 1284.60                                     | 1245.06                                        | 1179.86                        |
>
> | **Method**              | **ImageReward (T³Bench) $ \uparrow $** |
> | ----------------------- | -------------------------------------- |
> | SDS (~1h)               | -0.8838 $\pm$ 0.5142                   |
> | VSD (~5h)               | -0.6737 $\pm$ 0.4098                   |
> | **TraCe (ours, ~3.5h)** | -0.4533 $\pm$ 0.4635                   |

---

> ### Comment · Reviewer_XMDr · 2025-08-08
>
> Dear authors,
>
> I sincerely regret that it is impossible to see under the current rebuttal process guidelines the additional qualitative results that shows TraCe brings about more drastic improvement in generation quality, especially when it is applied on 2D SDS - the results of 2D image optimization of Figure 3 at Supplementary Materials, does not show that great of a difference in quality, other than that the saturation seems to be a bit lessened. Even the results at Figure 1, which the authors claim as "drastic", show marginal improvement over SDS / VSD baselines at best. I am stating this because I have seen this level of minor difference in texture and geometry occurring from simply selecting a different random seed.
>
> That being said, I am reminded that the theoretical basing for this paper is very sound and acceptable - I just think that this methodology has more potential to it, especially in terms of sheer performance improvement, as drastic as VSD / ISM had. The authors have also provided within a short while an additional evaluation result in implicit NeRF baseline, as I have requested. I sincerely appreciate it and the performance improvement shown in the metric. In this light, I am changing my mind more towards scoring for acceptance.
>
> However, before raising my score, I do want to question how TraCe performs under different CFG values. I ask this because CFG=20 is a very unusual value, not large as SDS's CFG=100 nor not nominal as VSD (and standard ancestral sampling's) CFG=7.5 value. The author's claim of using a "fair CFG value" with CFG=20 doesn't really land with me - is there a special reason you have selected this CFG value? How does TraCe perform in more standard CFG settings such as CFG=7.5 or CFG=100? I thank the authors for their detailed response.

---

> ### Author Response · Authors · 2025-08-08
> **Response**
>
> We sincerely thank the reviewer for their detailed consideration, positive re-evaluation of our work, and this insightful follow-up question. We are happy to clarify the choice of CFG value and our method's performance across different settings.
>
> ---
>
> Our **selection of CFG=20** for the main comparisons was intended to find a balanced setting that demonstrates TraCe's stability. TraCe's performance is remarkably stable across a wide range of CFG values from 15 to 100. CFG=20 represents the start of this stable plateau, showcasing that our method achieves high-fidelity results without resorting to the extreme CFG values (e.g., 100) required by standard SDS, while also maintaining quality at a moderate guidance scale where methods like SDS may not perform optimally.
>
> We also provide the newly conducted experiment with ImageReward Score. Please check the table below (and compare it with the table 2 in my initial response to [W2]). For TraCe at CFG=100, TraCe's performance is the **best among all compared methods**, demonstrating its robustness and ability to handle strong guidance. For TraCe at CFG=7.5, TraCe is highly competitive, performing **better than methods like SDS, VSD, etc**. While it scores only slightly lower than DreamerXL (by ~0.08 lower on ImageReward) and ISM (by ~0.02 lower on ImageReward) in this specific setting, it remains one of the top-performing methods.
>
> It is crucial to note that these strong and stable results were achieved using the default 3DGS implementation and hyper-parameters from the studio framework to ensure a fair and direct comparison which are not specifically pruned for our method. The fact that TraCe performs robustly across a wide range of CFG values without any specialized tuning speaks to the inherent stability of our Schrödinger Bridge-based optimization trajectory. **We believe its performance could be further enhanced with further pruned hyper-parameters for TraCe**.
>
> | Methods         | **ImageReward (T³Bench) $ \uparrow $** |
> | --------------- | -------------------------------------- |
> | TraCe (CFG=100) | -0.2241 $\pm$ 0.9835                   |
> | TraCe (CFG=75)  | -0.2108 $\pm$ 0.8849                   |
> | TraCe (CFG=50)  | -0.2090 $\pm$ 0.9727                   |
> | TraCe (CFG=30)  | -0.2264 $\pm$ 0.9722                   |
> | TraCe (CFG=20)  | -0.2855 $\pm$ 0.8909                   |
> | TraCe (CFG=10)  | -0.3420 $\pm$ 0.9909                   |
> | TraCe (CFG=7.5) | -0.4182 $\pm$ 0.8825                   |
> | TraCe (CFG=5)   | -0.5896 $\pm$ 0.9953                   |
>
> ---
>
> We also want to respectfully argue that TraCe's improvements differentiate other methods from simple random seed variation. We would like to guide the reviewer's attention to specific areas of improvement within the provided examples (we take VSD as an example, and the results are generated with the same seed):
>
> 1. A gothic cathedral..." in Figure 9 (supplementary): Please observe the correct shape of crucifix on the top of tall arches. Our result also shows sharper, more defined architectural lines and cleaner window details. The VSD version, in contrast, appears slightly blurrier, with the fine structural elements being less distinct.
>
> 2. "A sea turtle" in Figure 9 (supplementary): The VSD turtle has a smoother, less detailed surface and exhibits *an unexpected glow of light* on its tail—an artifact that is absent in our more realistically rendered version. VSD's "A... amigurumi motorcycle" also has this unexpected glow of light at the front of the motorcycle. The shell pattern on our turtle is also crisper and more intricate, and the skin texture is more pronounced.
>
> 3. "A stack of pancakes..." in Figure 9 (supplementary): In our result, the structure of the bottom part of the maple syrup is cleaner, and it has a more viscous, glistening appearance, and the texture of the individual pancakes is more defined. The VSD version's syrup and pancake texture appears somewhat flatter and less detailed.
>
> ---
>
> We agree wholeheartedly with your assessment that our methodology holds great potential. We believe our current results are the first step in unlocking that potential by demonstrating a better generation process.

---

> ### Author Response · Authors · 2025-08-09
> **Gentle Reminder**
>
> Dear Reviewer XMDr,
>
> We sincerely thank you again for your constructive and detailed engagement with our paper.
>
> We have posted a comprehensive response that we hope addresses all of your follow-up questions regarding the significance of our method's improvements and its performance across different CFG values. As the author-reviewer discussion period is approaching its end, we just wanted to send a brief and gentle reminder.
>
> We hope our detailed answer has helped clarify the final points. Regardless, we are very grateful for your time and the invaluable insights you have provided throughout this process.
>
> Best regards,
>
> The Author's of paper ID 4750

---

> > ### Comment · Reviewer_XMDr · 2025-08-09
> >
> > I thank the authors for their detailed analysis and elaboration on the paper. I raise my score towards acceptance.

---

> > > ### Author Response · Authors · 2025-08-09
> > > **Thank you**
> > >
> > > Dear Reviewer XMDr,
> > >
> > > We are delighted to read your final decision. Thank you so much for raising your score and for your thoughtful engagement with our work. Your constructive feedback was instrumental in helping us strengthen the paper, and we are truly grateful for your time, expertise, and support.
> > >
> > > Sincerely,
> > >
> > > The Authors of paper ID 4750

---

### Official Review · Reviewer_z2hB · 2025-07-03

**Clarity:** 2
**Significance:** 2
**Originality:** 3
**Rating:** 4
**Confidence:** 3

**Summary:**

This paper introduces Trajectory-Centric Distillation (TraCe), a new optimization framework for text-to-3D generation. The authors establish a theoretical connection between Score Distillation Sampling (SDS) and Schrödinger Bridge (SB) problems, showing SDS as a special case. They then leverage this connection to construct a mathematically principled diffusion bridge between the current rendering and a text-aligned target image. Using this bridge, they train a LoRA-adapted diffusion model to guide 3D optimization, demonstrating superior performance, especially at lower CFG values, compared to existing state-of-the-art methods like SDS, VSD, and SDI.

**Questions:**

1. It's unclear how significant the adaptation via LoRA is. How much of the improvement is from using the SB trajectory versus simply adapting the diffusion model?

2. The 2D experiments show blurry results. Could more analysis be added regarding this?

**Ethical Concerns:**

["NO or VERY MINOR ethics concerns only"]

**Final Justification:**

The paper presents a solid analysis connecting Score Distillation Sampling (SDS) to the Schrödinger Bridge framework and introduces Trajectory-Centric Distillation (TraCe), an optimization method for text-to-3D generation. The contributions are well-motivated and demonstrate good performance. After reviewing the questions raised by other reviewers and the authors’ detailed rebuttal, I have decided to maintain my recommendation as a borderline accept.

**Limitations:**

More description about the limitations should be added.

**Paper Formatting Concerns:**

No major formatting concerns.

**Quality:**

3

**Strengths And Weaknesses:**

Strengths:

1. Establishing that SDS is a special case of the Schrödinger Bridge can provide theoretical insights to the community.

2. Experiments (quantitative and qualitative) demonstrate that TraCe outperforms baseline methods in fidelity, detail, and texture quality, especially at low CFG values.

3. The paper includes qualitative ablations to show the effectiveness of each component.

Weaknesses:

1. The paper heavily relies on CLIP scores, which may not fully capture geometry quality or 3D consistency. Adding metrics like FID on rendered views and 3D metrics would strengthen the claims. Consider moving the human study results from the supplementary to the main manuscript.

2. A more comprehensive quantitative comparison with recent state-of-the-art 3D generation methods is needed.

3. The diversity of the generated objects appears limited and could be further improved.

---

> ### Author Rebuttal · Authors · 2025-07-31
>
> > **[W1] Additional Evaluation Metrics**
>
> We thank the reviewer for your insightful suggestions on strengthening our evaluation.
>
> We chose the CLIP score for our quantitative analysis to ensure our results are fairly compared with prominent works in the field, such as ProlificDreamer, DreamFlow, and Score Distillation via Reparametrized DDIM, which **all use it as a key metric**. Moreover, to address the exact concerns about geometry and 3D consistency, we supplemented this with a user study involving 60 participants referring to DreamFlow’s user study settings. This study directly evaluated perceptual qualities that quantitative metrics cannot, including 3D consistency, photorealism, and prompt fidelity. The results, detailed in the supplementary (Tables 1-3), show a clear user preference for TraCe. While metrics like FID can assess rendered image realism, they don't capture 3D consistency or text alignment, and most 3D metrics require ground-truth data that is not available for text-to-3D generation. At the same time, human perceptual evaluation is often more persuasive than automated metrics [20]. We believe our user study provides a more holistic and meaningful assessment of quality for this task.
>
> In response to reviewer’s suggestion, we have also conducted additional quantitative experiments (please refer to **the tables in Reviewer XMDr W2**). We selected two recognized and common evaluation metrics for the task of 3D asset generation: GPT-4o (with an implementation based on GPTEval3D) and ImageReward (with an implementation based on T³Bench). We would also like to respectfully refer the reviewer to the supplementary material, where extensive qualitative results (Figures 4, 7-12) further demonstrate the practical improvements in geometric detail and 3D coherence our method provides. We will follow your excellent suggestion to move the user study results into the main paper.
>
> > **[W2] Comparison with More SOTA Methods**
>
> We thank the reviewer for the feedback. For the comparison, we selected the five most commonly used and state-of-the-art methods from SDS-based works.
>
> In response to reviewer’s suggestion, we have now added new comparison methods: Consistent3D (2024) and DreamerXL (2024). **Please refer to the tables in Reviewer XMDr W2.** The new experiments evaluate TraCe using two state-of-the-art benchmarks: T³Bench, which is based on the ImageReward score, and GPTEval3D, which uses GPT-4o for a multi-faceted assessment. The results from both benchmarks demonstrate TraCe's superior performance.
> In T³Bench evaluation, a higher score (closer to zero) is better. TraCe achieves the highest score of -0.2855, significantly outperforming all other methods, including the new comparison methods DreamerXL (-0.3382) and consistency (-0.5497). This is achieved with a competitive optimization time of 14 minutes.
>
> GPTEval3D provides a more detailed breakdown. TraCe achieves the highest score across all six categories evaluated by GPT-4o: 3D Plausibility, Texture Details, Geometry Details, Text-Asset Alignment, Text-Geometry Alignment, and Overall. Notably, in the "Overall" assessment, TraCe scores 1028.033, showing a clear lead over other methods.
>
> > **[W3] Regarding Generation Diversity**
>
> Unlike other SDS-based methods, our method's diversity is not negatively impacted because we use a low CFG value.
>
> We agree this is an important aspect, and it is often linked to a trade-off with generation quality via the CFG value. It is well-known that high CFG values can reduce diversity by forcing the model into a narrow mode of the distribution. A key advantage of our method (TraCe) is its ability to achieve high-fidelity results at low and stable CFG settings (e.g., 15-20), which is better for preserving the natural diversity offered by the diffusion prior. While we do not claim that TraCe fundamentally solves the diversity challenge for SDS-based methods, its operational characteristics are more conducive to preserving variety compared to methods that rely on high CFG for quality.
>
> Our experiments in the supplementary material (Section H, Figure 5) demonstrate this capability. For prompts like "an amigurumi motorcycle" or "an overstuffed pastrami sandwich," TraCe produces multiple distinct 3D instances with clear variations in geometry, structure, and style. While we present this to show that our method generates varied and plausible assets , we also note that this level of diversity is in line with that of other leading score distillation methods.
>
> > **[Q1] Ablation Study on LoRA Adaptation**
>
> We thank the reviewer for this insightful question regarding the significance of the LoRA adaptation versus the Schrödinger Bridge (SB) trajectory. To precisely disentangle these effects, we have conducted a new quantitative ablation study, complementing the qualitative results in Figure 5. We evaluated different configurations of our method using ImageReward, and we used 43 prompts from DreamFusion online gallery [1] for generation.
>
> The results clearly show that both components contribute significantly and, more importantly, work synergistically to achieve the best performance (we keep other hyper-parameters the same for fair comparisons):
>
> | Method Configuration                      | ImageReward Score ($ \uparrow $) |
> | ----------------------------------------- | --------------------- |
> | LoRA off & scheduled t-sampling off       | -0.4488 $\pm$ 0.9964       |
> | LoRA off & scheduled t-sampling on        | -0.3389 $\pm$ 0.9721       |
> | LoRA on & scheduled t-sampling off        | -0.4020 $\pm$ 1.0019       |
> | LoRA on & scheduled t-sampling on (TraCe) | -0.2486 $\pm$ 0.8909       |
>
> Enabling the scheduled t-sampling alone provides a strong foundation, significantly improving the score from -0.4488 to -0.3389. This highlights the inherent benefit of optimizing along the more stable trajectory from source to target defined by our SB formulation. Similarly, adding LoRA adaptation in isolation also improves the baseline score, demonstrating that allowing the diffusion model to adapt is beneficial. Moreover, the full TraCe method, which combines both components, achieves the best score of -0.2486. This result is substantially better than applying either component individually, demonstrating a clear synergistic effect. This shows that the LoRA-adapted model performs best when it is trained to learn the specific score dynamics of the SB trajectory from source to target.
>
> [1] https://dreamfusion3d.github.io/gallery.html
>
> > **[Q2] Analysis of Blurry Results in 2D Experiments**
>
> Thank you.
>
> The 2D experiment presented in the supplementary material (Section F, Figure 3) is designed as a diagnostic tool to isolate and directly compare the behavior of different score distillation gradients, removing the complexities of 3D representations. This type of analysis, where a 2D image is optimized directly, has been used in prior work (e.g., VSD) to understand the inherent properties of the guidance signal itself.
>
> **The goal of this experiment is not to compete with state-of-the-art, multi-step 2D text-to-image samplers, but rather to demonstrate the *relative* improvement of the TraCe gradient** over other score distillation losses like SDS, CSD, and VSD. As shown in Figure 4 in main manuscript and Figures 4, 7-12 in the supplementary, our method consistently generates images with **enhanced visual fidelity, quality, and fewer artifacts compared to these other distillation techniques.** For example, the results from SDS often contain noise, CSD can produce overly stylized outputs, and VSD can lack sharpness. Our method mitigates these issues, yielding more realistic textures and clearer details.
>
> This analysis provides evidence that our TraCe framework offers a fundamentally more stable and accurate guidance signal. This improvement in the 2D domain directly translates to the superior quality and reduced artifacts we observe in the more complex task of text-to-3D generation.

---

> > ### Comment · Reviewer_z2hB · 2025-08-05
> >
> > Thank you to the authors for the thorough and well-organized rebuttal. The additional experiments, clarifications, and newly added comparisons effectively address my concerns. As my initial evaluation was already positive, I will maintain my current score.

---

> > > ### Author Response · Authors · 2025-08-05
> > > **Response**
> > >
> > > Dear Reviewer z2hB,
> > >
> > > Thank you for your positive feedback and for confirming that our rebuttal addressed your concerns. We are very grateful for your thoughtful comments, which have helped us improve the paper.
> > >
> > > Thank you again for your time and engagement.
> > >
> > > The Authors of paper ID 4750.

---

### Note · Authors · 2025-08-13

Dear Area Chair dmgH and reviewers,

We sincerely thank the Area Chair and all reviewers for their invaluable time and constructive feedback. We are very grateful for the opportunity to engage in a productive discussion during the rebuttal period.

We are particularly encouraged that our rebuttal was received so positively, leading three reviewers to state their intention to raise their scores. We are grateful that they subsequently increased their scores from 3 (borderline reject) to (at least) 4 (borderline accept). This resulted in a unanimous score of (>=) 4 across the panel, reflecting a consensus that the initial concerns were successfully resolved.

During the discussion, we clarified the significance of our method's improvements over the SOTA methods. We are confident that our work makes a solid contribution by introducing a more principled and stable optimization trajectory for text-to-3D generation. We are also fully committed to incorporating all discussed clarifications and new results into the final camera-ready version to further strengthen the paper.

Thank you once again for your careful consideration and for the opportunity to strengthen our paper through this interactive process.

Sincerely,

The Authors of Paper ID 4750

---

### Decision · Program_Chairs · 2025-09-17

**Decision:**

Accept (poster)

**Comment:**

The authors propose a new “generative” approach to transferring “text to 3D”. Their contribution is two-fold and proceeds with establishing an equivalence between the “text-to -3D” generation and the Schrodinger bridge in stochastic diffusion, followed with an extensive experimental validation, including numerous experiments requested (at times quite late) by reviewers. The question and rebuttal period has been, by many measures, thorough and informative given some of the reviewers’ pertinent comments and the prompt and equally conscientious responses by the authors.
Specifically, they formulate the data generation process as that of learning an optimal transport problem between a source and a sought target distribution, thereby enabling high-quality generation and independence of Classifier-free Guidance (CFG) values. The authors first establish an SDS as a simplified instance of the Schrödinger Bridge framework, so that it effectively exploits the reverse process of a Schrödinger Bridge, which, under specific conditions (e.g., a Gaussian noise as one end), collapses to SDS's score function of the pre-trained diffusion model. A Trajectory-Centric Distillation (TraCe) is subsequently introduced as text-to-3D generation approach.
 This paper had a long, extended and quite thorough exchange between the authors and the reviewers. The reviewers all agreed that the paper was theoretically solid and varied on the degree to which they thought the experiments validated the claims. Three of the reviewers who had an initially lower ranking, raised it upon their admission that the authors had been very vigilant, cooperative in punctually in answering all questions as well as running some requested experiments in a rather surprisingly timely fashion. All reviewers agreed that the contribution was theoretically solid but would have benefitted from additional experiments some of which had been run as a response.  They capped their scores borderline accept, still insisting more extensive experiments in addition to the included ablation studies and all other additional experiments run during the question-answer period.
As an AC and having followed the back-and-forth discussion, believe that the paper has a contribution of merit, and that the authors have navigated the conceptual questions and admirably quickly have answered questions and delivered on the additional experiments they were requested to produce.  In light of this exchange, and of  the AC's sollicitation of a discussion  to potentially provide any strong leaning ,    and of the AC's own quick reading of the paper, the AC supports the acceptance of this paper.